# LEARNING LABEL DISTRIBUTION WITH SUBTASKS

## ABSTRACT

Label distribution learning (LDL) is a novel learning paradigm that emulates label polysemy by assigning label distributions over the label space. However, recent LDL work seems to exhibit a notable contradiction: 1) some existing LDL methods employ auxiliary tasks to enhance performance, which narrows their focus to specific domains, thereby lacking generalization capability; 2) conversely, LDL methods without auxiliary tasks rely on losses tailored solely to label distributions of the primary task, lacking additional supervised information to guide the learning process. In this paper, we propose $\mathcal{S}$-LDL, a novel and minimalist solution that partitions the label distribution of the primary task into subtask label distributions, i.e., a form of pseudo-supervised information, to reconcile the above contradiction. $\mathcal{S}$-LDL encompasses two key aspects: 1) an algorithm capable of generating subtasks without any extra knowledge, with subtasks deemed valid and reconstructable via our analysis; and 2) a plug-and-play framework seamlessly compatible with existing LDL methods, and even adaptable to derivative tasks of LDL. Experiments demonstrate that $\mathcal{S}$-LDL is effective and efficient. To the best of our knowledge, this represents the first endeavor to address LDL via subtasks. The code will soon be available on GitHub to facilitate reproducible research.

## 1 INTRODUCTION

Multi-label learning (MLL) (Zhang and Zhou, 2013) handles label polysemy in a binary manner, whereas label distribution learning (LDL) (Geng, 2016) offers a more nuanced perspective by answering: "*How much does each label $y$ describe the instance $\boldsymbol{x}$?*". This is accomplished through the concept of a *label distribution $\boldsymbol{d}$*, which is a form of probability simplex that assigns a real value (i.e., *description degree $d_{\boldsymbol{x}}^y$*) to each label of each instance. This form introduces a quantitative manner to address label polysemy and extends LDL's practical applications to a wider range, e.g., counting (or grading) (Geng et al., 2013; Wu et al., 2019), sentiment analysis (Chen et al., 2020; Le et al., 2023), segmentation (Gao et al., 2017; Li et al., 2023b), etc. Concurrently, more and more derivative tasks of LDL (González et al., 2021b; Lu and Jia, 2022; Wang, Jing and Geng, Xin, 2019; Xu and Zhou, 2017; Xu et al., 2019) are emerging to offer assistance in various real-world dilemmas.

However, LDL encounters a spectrum of challenges: 1) label distributions are bound by two constraints, non-negativity (i.e., $d_{\boldsymbol{x}}^y \geq 0$) and sum-to-one (i.e., $\sum_{y \in \mathcal{Y}} d_{\boldsymbol{x}}^y = 1$), and are often formed from mixture distributions, posing significant hurdles for fitting, particularly when employing a maximum entropy model (Shen et al., 2017); 2) label distribution matrices are usually obtained via crowdsourcing, which is time-consuming and labor-intensive, so one often copes with scarce and low-quality datasets (Wang et al., 2023). These two key issues stand as formidable barriers to performance improvement in LDL.

With the widespread use of multi-task learning, some LDL work tries to compensate for performance from the perspective of *auxiliary tasks*, which are learned concurrently alongside the primary task, thereby refining its representations and ultimately boosting performance. Unfortunately, though these methods can exploit additional supervised information, they 1) do not address the first key issue mentioned above; and 2) require extra knowledge (e.g., facial characteristics (Chen et al., 2020), pathology criteria (Wu et al., 2019), emotion wheel theory in psychology (Yang et al., 2017a), etc.) or similar domain-specific data (Zhao et al., 2023b), limiting their generalization capability to those corresponding specific domains. Conversely, LDL methods that do not take advantage of auxiliary tasks, despite their efforts in loss function engineering and network structure design, they 1) do not address the second key issue mentioned above; and 2) focus solely on one aspect of label correlations

(e.g., correlation of local instances (Jia et al., 2019), ranking relation (Jia et al., 2023), suboptimal label (Wang, Jing and Geng, Xin, 2019), etc.), each with its own set of limitations.

The generalizability across various domains appears to conflict with the ability to exploit additional data, so benefiting from both simultaneously seems elusive. However, we can still see the light from some MLL methods, which partition the label space and apply operations on these subspaces (Tsoumakas et al., 2008; 2010). These methods construct *subtasks* without involving extra knowledge and exhibit applicability across various domains. Intuitively, in the context of LDL, reliable supervised information can be generated from these subtasks, which can eventually be aggregated and reconstructed to the information of the primary task via ensemble strategies. Although existing label distribution ensemble practices demonstrate promising performance (González et al., 2021a; Shen et al., 2017), they focus only on the supervised information of the primary task.

In this paper, we introduce $\mathcal{S}$-LDL, a novel and minimalist label distribution learning algorithm that constructs and exploits subtasks, to reconcile the contradiction between the generalizability across various domains and the ability to exploit additional data. Serving as auxiliary tasks, subtasks 1) provide different views of the primary task distribution, rendering the mixture of distributions more traceable (i.e., the key issue one); 2) furnish additional supervised data to mitigate the scarcity and ambiguity inherent in LDL datasets (i.e., the key issue two); 3) require no extra knowledge from specific domains; and 4) emphasize various label correlations via partitioning of the label space.

The main contributions of this paper are outlined below: 1) we propose $\mathcal{S}$-LDL, which is considered the first endeavor to address LDL via subtasks; 2) our analysis shows the validity and reconstructability of these subtasks; 3) we present a plug-and-play framework seamlessly compatible with existing LDL methods, and adaptable to derivative tasks of LDL; and 4) the code will be available on GitHub soon, facilitating reproducible research endeavors.

## 2 RELATED WORK

**LDL**   Our work is mainly related to LDL. Initially employed to tackle age estimation (Geng et al., 2013), LDL has evolved into a novel machine learning paradigm (Geng, 2016), which is supported by theoretical underpinnings (Wang and Geng, 2019) and features various derivative tasks (González et al., 2021b; Lu and Jia, 2022; Wang, Jing and Geng, Xin, 2019; Xu and Zhou, 2017; Xu et al., 2019). Most methods focus on improving performance via loss function engineering (Jia et al., 2019; 2023; Ren et al., 2019; Wen et al., 2023) or efficient model structures (González et al., 2021a; Jin et al., 2024; Shen et al., 2017; Yang et al., 2017b), while some work is dedicated to practical application scenarios (Gao et al., 2017; Li et al., 2023a; Shirani et al., 2019; Wu et al., 2019). However, the scarcity of label distribution datasets and the complexity of the label distribution itself make it difficult to further improve performance, at which point one may think of leveraging auxiliary tasks.

**LDL with auxiliary tasks**   While there are LDL methods that leverage auxiliary tasks to enhance performance, they often rely on knowledge from disparate domains, extending beyond the scope of the LDL task. For example, LDL-ALSG (Chen et al., 2020) designs auxiliary tasks dedicated to facial emotion recognition, necessitating the use of external tools to extract facial points and action units from human faces. Wu et al. (2019) exploit the Hayashi criterion, a rule for counting and grading in acne lesions, which results in their method being only applicable in a small branch of the dermatology field. Yang et al. (2017a) employ a multi-task framework for image emotion classification, designing constraints inspired by Mikel's wheel, a psychological emotion model, which also suffers from similar limitations. As LDL methods of transfer learning, GLDL (Zhao et al., 2023b) utilizes data from one or more source domains, which is not easy to obtain in practical applications. The need for specific extra knowledge significantly narrows the application scenarios of these methods.

**MLL with partitioning of the label space**   For reference, there exist MLL methods based on partitioning of the label space, which can construct multi-label subtasks without involving additional knowledge and can be widely used in various domains. The most classic related work is that of HOMER (Tsoumakas et al., 2008) and RA$k$EL (Tsoumakas et al., 2010), the former forms a hierarchy of label subspaces while the latter randomly selects label subspaces. Many subsequent papers have been inspired by them (Prabhu et al., 2018; Read et al., 2013; Wang et al., 2021). Read et al. (2014) present a general framework of label subspaces and provide some theoretical justification for it. Since

Table 1: Key notation and terminology in this paper

| Symbol | Description | Example |
|---|---|---|
| $\mathcal{Y} = \{y_j\}_{j=1}^L$ | Label space ($L$ labels) | $\mathcal{Y} = \{$HA, SA, SU, AN, DI, FE$\}$[1] |
| $\mathcal{Y}^\circ$ | Subtask label spaces | $\{\cdots, \mathcal{Y}^{(t)} = \{$HA, SA, SU, FE$\}, \cdots\}$ |
| $d_{\boldsymbol{x}_i}^{y_j}$ | Description degree of $\boldsymbol{x}_i$ about $y_j$ | $d_{\boldsymbol{x}_i}^{y_0} = 0.4$, i.e., HA describes $\boldsymbol{x}_i$ by 0.4 |
| $\boldsymbol{d}_i = (d_{\boldsymbol{x}_i}^{y_j})_{j=1}^L$ | Label distribution of $\boldsymbol{x}_i$ | $\boldsymbol{d}_i = (0.4, 0.05, 0.3, 0.1, 0.1, 0.05)$ |
| $\boldsymbol{D} = (\boldsymbol{d}_i)_{i=1}^N = (\boldsymbol{d}_{\bullet j})_{j=1}^L$ | Distribution matrix ($N$ samples) | $(\cdots, \boldsymbol{d}_i, \cdots)$ |
| $\boldsymbol{d}_i^{(t)} = (d_{\boldsymbol{x}_i}^{(t)y_j})_{j=1}^{\|\mathcal{Y}^{(t)}\|}$ | Subtask label distribution | $\boldsymbol{d}_i^{(t)} = (0.5, 0.0625, 0.375, 0.0625)$ |
| $\mathcal{D}^\circ$ | Subtask distribution matrices | $\{\cdots, \boldsymbol{D}^{(t)}, \cdots\}$ |
| $\boldsymbol{M} = (\boldsymbol{m}_t)_{t=1}^T = (M_{tj})$ | Mask matrix ($T$ anticipated tasks) | $(\cdots, \boldsymbol{m}_t = (1, 1, 1, 0, 0, 1), \cdots)$ |

label distribution contains rich knowledge, we can follow the patterns of these methods to construct label distribution subtasks.

**LDL with ensemble strategy** It is imperative to aggregate the output of subtasks. Fortunately, ensemble-based LDL methods have demonstrated promising performance. For instance, LDLFs (Shen et al., 2017) learns different label distributions on the leaf nodes of differentiable decision trees and learns weights that aggregate these label distributions. DF-LDL (González et al., 2021a) aggregates the label distribution of output of multiple base models by simple averaging, while Zhai et al. (2018) focus on aggregating the results of various neural networks via a combining learner. However, 1) the above methods are not suitable for incomplete label spaces (i.e., subtask label spaces); and 2) *none* of them involve the partitioning of the label space, therefore no extra supervised information of label distributions is constructed.

Drawing from the analysis of the aforementioned related work, we introduce $\mathcal{S}$-LDL, which leverages pseudo-supervised information from subtasks to eliminate reliance on additional knowledge from disparate domains, and facilitates the creation of a novel knowledge dimension in a generic framework.

# 3 SUBTASK CONSTRUCTION

## 3.1 PRELIMINARY

**Notation** Vectors are denoted by lowercase bold letters, e.g., $\boldsymbol{v}$, and the corresponding regular letter with subscript $i$, i.e., $v_i$, indicates its $i$-th element. Matrices are denoted by uppercase bold letters, e.g., $\boldsymbol{A}$. The row vector $\boldsymbol{a}_i$ indicates its $i$-th row and the column vector $\boldsymbol{a}_{\bullet j}$ indicates its $j$-th column. $A_{ij}$ is the element in $i$-th row and $j$-th column of $\boldsymbol{A}$. The superscript $(t)$ indicates that a symbol corresponds to the $t$-th subtask. Table 1 outlines the key notation in this paper.

**Problem definition** Let $\boldsymbol{x} \in \mathbb{R}^P$ denote the feature of the instance and $\boldsymbol{d} \in \Delta^{L-1}$ denote the label distribution, where $\Delta^{k-1} \triangleq \{\boldsymbol{v} \in \mathbb{R}^k \,|\, \boldsymbol{1}\boldsymbol{v}^{\mathrm{T}} = 1, \boldsymbol{v} \geq 0\}$ is the $(k-1)$-dimensional probability simplex. The goal of LDL is to find a mapping $\zeta : \boldsymbol{x} \mapsto \boldsymbol{d}$. In this paper, we partition the label space $\mathcal{Y}$ corresponding to $\boldsymbol{d}$ to obtain the subtask label space set $\mathcal{Y}^\circ$, then accordingly generate pseudo-supervised information, i.e., subtask distribution matrix set $\mathcal{D}^\circ$, to guide the learning of $\zeta$.

**Technical challenges** Our first challenge arises from *the exponential growth in partitions* as the number of labels increases (Tsoumakas et al., 2010). When generating $T$ tasks from a label space with $L$ labels, the number of unique partitions is given by $(2^L - L - 2)! / (T!(2^L - L - 2 - T)!)$. This makes it impractical to calculate metric for each case to select subtasks. We tackle this challenge in a mask matrix learning manner. The second challenge lies in *discerning reasonable partitions*. Since the label distribution matrix is usually imbalanced in average description degree (Zhao et al., 2023a), some partitions exhibit unreasonable local ignorance. As a result, the corresponding spaces struggle to handle the majority of instances, because 1) theoretically, there is no objective standard for the degree of negative correlation; and 2) empirically, weakly or negatively correlated information is

---

[1] HA, SA, SU, AN, DI, FE, and NE represent the seven common emotions in sentiment analysis datasets, namely `happiness`, `sadness`, `surprise`, `anger`, `disgust`, `fear`, and `neutral`, respectively.

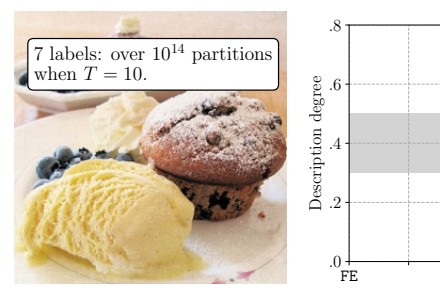 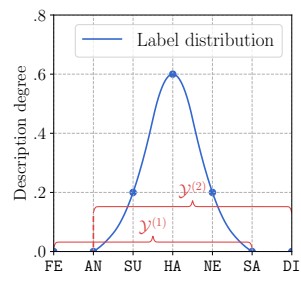

   (a) Huge number of partitions   (b) Unreasonable ignorance   (c) Lacking new knowledge [2]

Figure 1: **(a)** is sourced from the `emotion6` (Yang et al., 2017b) dataset, which has only 7 labels, but the number of potential partitions is huge. **(b)** exemplifies a subtask label space {FE, AN, DI}, which is challenging to describe (a). **(c)**'s two subtask label spaces encompass all descriptive information, meaning no new knowledge is generated about (a). This example vividly illustrates the limitations that may result from local ignorance and lack of diversity in subtask label spaces.

easily overlooked by human annotators in crowdsourced datasets. To mitigate this, we incorporate the description degree as a metric for the reliability of supervised information in guiding the generation of subtask masks. The third challenge is *avoiding analogous pseudo-supervised information*, i.e., to generate label distributions containing new knowledge (González et al., 2021b). This necessitates fostering richness and diversity in both subtask label spaces and distributions. To achieve this objective, we 1) minimize pairwise similarity among subtask masks; and 2) normalize each subtask label distribution to yield brand new insights distinct from the label distribution of the primary task. Fig. 1 portrays an illustrative example of these challenges.

### 3.2 LEARNING SUBTASK MASKS

Let $\boldsymbol{M} \in \{0, 1\}^{T \times L}$ denote the subtask mask matrix, where $T$ represents the number of anticipated tasks. To ensure that the subtask label spaces contain as reliable information as possible, the learning of the subtask mask matrix can be converted into this problem: $\arg\max_{\boldsymbol{M}} \|\boldsymbol{D}\boldsymbol{M}^\top\|_{\mathrm{F}}$.

Obviously, a senseless solution is $\boldsymbol{m}_t = \boldsymbol{1}$ where $t = 1, \cdots, T$, i.e., all pseudo-supervised information is equivalent to the primary task information. Therefore, solving the above problem alone is inappropriate. To address this, we consider pairwise similarity among subtask masks. We also employ exponential tricks to convert maximization into minimization. Finally, $\boldsymbol{M}$ is calculated as

$$\boldsymbol{M}^* = \arg\min_{\boldsymbol{M}} \left( \frac{1}{NT} \sum_{t=1}^{T} \sum_{i=1}^{N} \exp\left(-\boldsymbol{d}_i \boldsymbol{m}_t^\top\right) + \frac{2\lambda}{(T(T-1))} \sum_{i, j, i \neq j} \frac{\boldsymbol{m}_i \boldsymbol{m}_j^\top}{\|\boldsymbol{m}_i\| \|\boldsymbol{m}_j\|} \right), \quad (1)$$

$$\text{s.t. } M_{tj} \in \{0, 1\}; \, t = 1, \cdots, T; \, j = 1, \cdots, L,$$

where $\lambda$ is a trade-off parameter. Eq. (1) is slightly more complicated than conventional integer programming. For convenience, we solve it using the stochastic gradient descent (SGD) method, with its constraint enforced via sigmoid (a conversion threshold is set, where outputs greater than it are set to 1, while those below are set to 0). Refer to Section 4.1 for an analysis of the validity of Eq. (1).

### 3.3 GENERATING SUBTASK DISTRIBUTIONS

We slice the label distribution matrix according to the subtask label space. To generate diversified subtask label distributions, we perform normalization on each subtask distribution with

$$[\mathcal{N}_{\mathrm{SUM}}(\boldsymbol{v})]_j = \frac{v_j}{\sum_{i=1}^{|\boldsymbol{v}|} v_i}. \quad (2)$$

---

[2]Despite the discrete label space, in the field of LDL, the label distribution is intentionally plotted as a curve, to distinguish it from the logical labels.

---

**Algorithm 1** Subtask construction

---

**Input**: Input matrix $\boldsymbol{D}$, trade-off parameter $\lambda$, anticipated number of subtasks $T$.
**Output**: Subtask distribution matrices $\mathcal{D}^\circ$ (with corresponding subtask label spaces $\mathcal{Y}^\circ$).
1: Initialization: $\mathcal{Y}^\circ = \{\varnothing\}, \mathcal{D}^\circ = \{\varnothing\}$;
2: Calculate $\boldsymbol{M}$ using SGD;            ▷ (Eq. (1))
3: **for** $t = 1$ **to** $T$ **do**
4:   $\mathcal{Y}^{(t)} \leftarrow \{y_j\}$ **if** $M_{tj} = 1$;
5:   **if** $|\mathcal{Y}^{(t)}| = L$ **or** $|\mathcal{Y}^{(t)}| \leq 1$ **then**
6:    **continue**; /* Ignore invalid subtask masks. */
7:   **end if**
8:   /* The clip$(x, a, b)$ function limits $x$ to be within $[a, b]$. */
9:   $\boldsymbol{D}^{(t)} \leftarrow \mathrm{clip}(\boldsymbol{d}_{\bullet j}, \varepsilon, 1)$ **if** $y_j \in \mathcal{Y}^{(t)}$; /* $\varepsilon$ is a very small positive number. */
10:   **for** $i = 1$ **to** $N$ **do**
11:    Normalization: $\boldsymbol{d}_i^{(t)} \leftarrow \mathcal{N}_{\mathrm{SUM}}(\boldsymbol{d}_i^{(t)})$;       ▷ (Eq. (2))
12:   **end for**
13:   $\mathcal{Y}^\circ = \mathcal{Y}^{(t)} \cup \mathcal{Y}^\circ, \mathcal{D}^\circ = \boldsymbol{D}^{(t)} \cup \mathcal{D}^\circ$;
14: **end for**

---

**Algorithm 2** $\mathcal{S}$-LDL (shallow regime)

---

**Input**: Feature matrix $\boldsymbol{X}$, label distribution matrix $\boldsymbol{D}$, testing instance $\boldsymbol{x}'$.
**Output**: Predicted label distribution $\boldsymbol{d}'$ for instance $\boldsymbol{x}'$.
1: Initialize parameter of each estimator;
2: $\mathcal{D}^\circ \leftarrow \mathrm{SC}(\boldsymbol{D})$;
3: **for** $t = 1$ **to** $|\mathcal{D}^\circ|$ **do**
4:   Fit an estimator $f^{(t)}$ on dataset $\{\boldsymbol{X}, \boldsymbol{D}^{(t)}\}$;
5:   $\boldsymbol{d}^{(t)\prime} \leftarrow f^{(t)}(\boldsymbol{x}')$;
6: **end for**
7: Concatenate $\boldsymbol{X}$ and all of the $\boldsymbol{D}^{(t)}$s to get $\boldsymbol{Z}$, where $t = 1, \cdots, |\mathcal{D}^\circ|$;
8: Fit an estimator $f$ on dataset $\{\boldsymbol{Z}, \boldsymbol{D}\}$;
9: Concatenate $\boldsymbol{x}'$ and all of the $\boldsymbol{d}^{(t)\prime}$s to get $\boldsymbol{z}'$, where $t = 1, \cdots, |\mathcal{D}^\circ|$;
10: $\boldsymbol{d}' \leftarrow f(\boldsymbol{z}')$;

---

The rationale for utilizing $\mathcal{N}_{\mathrm{SUM}}$ as the normalization function can be found in Section 4.2. The overall subtask construction process, denoted by SC, is illustrated in Alg. 1. Then, one can naturally come up with an adaptive LDL pipeline based on the shallow regime, as depicted in Alg. 2.

## 4   ANALYSIS ABOUT SUBTASK CONSTRUCTION

In this section, we analyze the subtask construction algorithm SC by studying the following questions:

- $\mathcal{Q}_1$: Are the subtask spaces provided by Eq. (1) valid for performance improvement? Can one configure $\lambda$ and $T$ in Eq. (1) without any prior knowledge?
- $\mathcal{Q}_2$: Are the subtask label distributions provided by Eq. (2) reconstructable? Can one replace Eq. (2) with other normalization functions?
- $\mathcal{Q}_3$: What is the overall time complexity of SC? Is it practical for large-scale datasets?

The validity, reconstructability, and complexity analysis are conducted for $\mathcal{Q}_1$, $\mathcal{Q}_2$, and $\mathcal{Q}_3$, respectively.

### 4.1   VALIDITY ANALYSIS

Eq. (1) manages the intricate task of selecting subtask spaces via $\lambda$ and $T$. On the one hand, we strive to explain that it is useful for performance improvement to suppress local ignorance and increase diversity of each subtask label space simultaneously. On the other hand, we seek to determine the appropriate $\lambda$ and $T$ without any prior knowledge. To this end, we design the following two metrics.

**Definition 1** (Information rate)**.** *We call it informative if $M_{tj} = 1$ where $t = 1, \cdots, T$ and $j = 1, \cdots, L$. Let $I$ be the summation of information; we define the information rate as $I/(TL)$.*

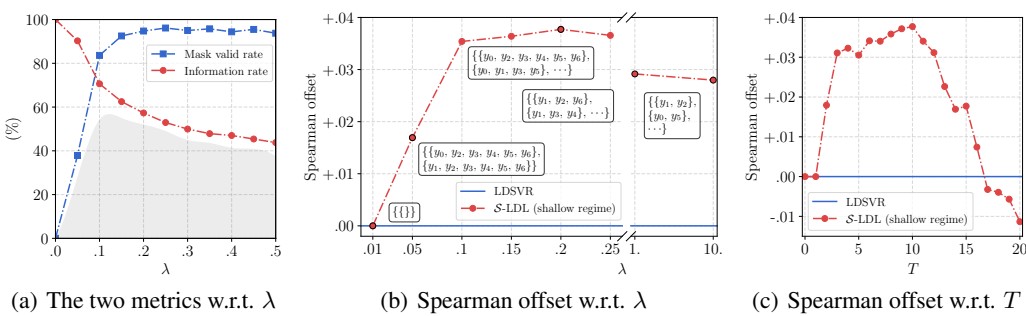

Figure 2: Visualized results of the validity analysis. Results of **(a)** are the average of experiments on all datasets (introduced in the appendix), while results of **(b)** and **(c)** are on the `emotion6` dataset. The blue lines in (b) and (c) represent performance without auxiliary tasks.

**Definition 2** (Mask valid rate). *Let $\delta(\cdot, \cdot)$ be the Kronecker delta function. For all $t = 1, \cdots, T$, the following are considered counting of invalid masks: 1) $\delta(\boldsymbol{m}_t, \boldsymbol{1})$, or 2) $\delta(|\boldsymbol{m}_t|!, 1)$, or, 3) excluding masks in cases 1) and 2), for any remaining mask index $i$, $\delta(\boldsymbol{m}_i, \boldsymbol{m}_j)$, where $j = 1, \cdots, i$.[3] Let $S$ be the summation of all invalid subtasks; we define the mask valid rate as $1 - S/T$.*

While it is unknown which metric is more important, we intuitively claim that higher values for both metrics are likely to lead to better performance. First, we calculate the average of these two metrics for all datasets with varying $\lambda$, results of which are shown in Fig. 2(a). The grey area depicts the average of the two metrics, implying that the appropriate value of $\lambda$ may be greater than 0.1.

It is important to highlight that Alg. 2 relies on naive concatenation operations and is not tied to representation learning. Consequently, any performance improvement over the base estimator is solely attributed to the effects of the subtask label distributions. Therefore, we employ Alg. 2 as the "scaffolding" for our analysis. With $\lambda$ varying and $T$ fixed at 10, we conduct ten-fold experiments repeated 10 times on the `emotion6` dataset using Alg. 2. Here, $f^{(t)}$s and $f$ are implemented by a representative LDL method, LDSVR (Geng and Hou, 2015). We record the average Spearman's coefficient (the higher the better). The results, which are shown in Fig. 2(b), support our claim. The panels from left to right display examples of subtask label spaces when the $\lambda$ is 0.01, 0.05, 0.2, 1, and 10, respectively. When $\lambda$ is suitable, label spaces are diverse and do not have excessive local ignorance; as $\lambda$ decreases, label spaces tend to be homogeneous, and invalid masks account for the majority; as $\lambda$ increases, the local ignorance of each label space becomes significant.

Besides, we also study the parameter sensitivity of $T$ with $\lambda$ fixed at 0.2. Results are shown in Fig. 2(c), illustrating that having a plethora of auxiliary tasks are detrimental to performance, which may be due to overfitting.

The validity analysis demonstrates that simultaneously avoiding local ignorance and homogeneity can lead to more efficient subtask label spaces, thereby improving performance. Without any prior knowledge, $\lambda$ and $T$ are recommended to be set to 0.2 and 10, respectively. Since the validity of Eq. (1) is ensured, one may wonder about the rationality and necessity of Eq. (2).

### 4.2 RECONSTRUCTABILITY ANALYSIS

We strive to choose a normalization function so that subtask label distributions retain more information, even efficacious enough to reconstruct the label distribution of the primary task. Theorem 1 illustrates that Eq. (2) is the only possibility.

**Theorem 1.** *Let each subtask label space form a connected graph with its each label as a node. Then merge these graphs according to their respective labels to form $\mathcal{G}$. If and only if $\mathcal{N}_{SUM}$ is used for normalization, the primary label distribution can be reconstructed from these subtask label distributions, when the following conditions are satisfied: 1) $\mathcal{G}$ is connected; 2) $\mathcal{G}$ covers all labels in the label space, and 3) corresponding description degrees of all cut vertices of $\mathcal{G}$ are not zero.*

---

[3]These three cases correspond to 1) masks that are exactly the same as the primary task; 2) masks that fail to form label distributions; and 3) duplicate masks among the remaining masks, respectively.

*Proof.* We solely discuss the extreme case where two subtask label spaces overlap with just one label. Further specialized cases can be deduced by the reader via induction. With a little bit of symbol abuse, let the general normalization function be defined as $\mathcal{N}(\boldsymbol{v}) \triangleq p(\boldsymbol{v})/q(\boldsymbol{v})$. Assume that there is a label distribution $\boldsymbol{d} = (d_1, \cdots, d_L)$ and its corresponding label space is $\mathcal{Y} = \{y_1, \cdots, y_L\}$. The two decompositions of $\mathcal{Y}$ are $\mathcal{Y}_{\boldsymbol{a}} = \{y_1, \cdots, y_k\}$ and $\mathcal{Y}_{\boldsymbol{b}} = \{y_k, \cdots, y_L\}$, respectively. It is obvious that $\mathcal{Y}_{\boldsymbol{a}} \cup \mathcal{Y}_{\boldsymbol{b}} = \mathcal{Y}$ and $\mathcal{Y}_{\boldsymbol{a}} \cap \mathcal{Y}_{\boldsymbol{b}} = \{y_k\}$. Let the subspace label distribution corresponding to these two decompositions be $\boldsymbol{a} = (a_1, \cdots, a_k)$ and $\boldsymbol{b} = (b_k, \cdots, b_L)$. According to our assumptions, $d_k \neq 0$. Then, for any integer $j \in [1, k]$, we have

$$\frac{a_j}{a_k} = \frac{[\mathcal{N}(\boldsymbol{d})]_j}{[\mathcal{N}(\boldsymbol{d})]_k} = \frac{[p(\boldsymbol{d})]_j}{[q(\boldsymbol{d})]_j} \frac{[q(\boldsymbol{d})]_k}{[p(\boldsymbol{d})]_k}. \tag{3}$$

Typically, for most normalization functions, $q(\cdot)$ is a normalizing constant, i.e., $[q(\boldsymbol{d})]_j = [q(\boldsymbol{d})]_k$. Thus Eq. (3) can be rewritten into $a_j[p(\boldsymbol{d})]_k = a_k[p(\boldsymbol{d})]_j$. Plug it into $\sum_{j=1}^{k} a_j = 1$, and do the same for $\boldsymbol{b}$ as well, and get

$$\frac{a_k \sum_{j=1}^{k}[p(\boldsymbol{d})]_j}{[p(\boldsymbol{d})]_k} = 1, \quad \frac{b_k \sum_{j=k}^{L}[p(\boldsymbol{d})]_j}{[p(\boldsymbol{d})]_k} = 1. \tag{4}$$

Add these two equations together, we have

$$[p(\boldsymbol{d})]_k + \sum_{j=1}^{L}[p(\boldsymbol{d})]_j = \frac{[p(\boldsymbol{d})]_k}{a_k} + \frac{[p(\boldsymbol{d})]_k}{b_k}. \tag{5}$$

Eq. (5) implies that $\sum_{j=1}^{L}[p(\boldsymbol{d})]_j$ must be given, and $[p(\boldsymbol{d})]_k$ is related to $d_k$, and only $d_k$. To make it possible, the only thing we can exploit is the sum-to-one constraint of $\boldsymbol{d}$, i.e., $\sum_{j=1}^{L} d_j = 1$. Therefore $[p(\boldsymbol{v})]_j = v_j$. Since $\sum_{i}^{|\boldsymbol{v}|}[\mathcal{N}(\boldsymbol{v})]_i = 1$, we have $q(\boldsymbol{v}) = \sum_{i}^{|\boldsymbol{v}|} v_i$, i.e., the finally deduced normalization function is Eq. (2). In this case, for any integer $j \in [1, L]$, we have

$$d_j = \begin{cases} \dfrac{a_j b_k}{a_k + b_k - a_k b_k}, & j = 1, \cdots, k \\ \dfrac{a_k b_j}{a_k + b_k - a_k b_k}, & j = k+1, \cdots, L \end{cases}, \tag{6}$$

which illustrates that the original label distribution $\boldsymbol{d}$ can be reconstructed by subtask label distributions $\boldsymbol{a}$ and $\boldsymbol{b}$. This is possible thanks to the use of $\mathcal{N}_{\text{SUM}}$. $\qquad\square$

Theorem 1 also states that it is not appropriate to replace Eq. (2) with the min-max or softmax function because doing so destroys the reconstruction information.

### 4.3 COMPLEXITY ANALYSIS

The overall time cost of SC is primarily influenced by the calculation of $\boldsymbol{M}$ and the normalization process. The time complexity of computing and updating $\boldsymbol{M}$ are $\mathcal{O}(L(TN + T^2))$ and $\mathcal{O}(LT)$, respectively. The time complexity of the normalization process is $\mathcal{O}(LTN)$. The

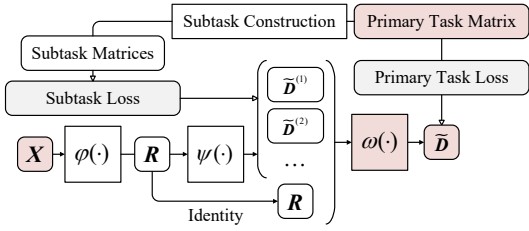

Figure 3: The overview of $\mathcal{S}$-LDL (deep regime). White, red and gray highlight our proposed, existing methods, and loss functions, respectively.

overall time complexity of each iteration of SC is $\mathcal{O}(L(TN + T^2))$, which is linear with respect to the number of instances and labels. Therefore, it is clear that SC can be applied to large-scale datasets.

## 5 $\mathcal{S}$-LDL OF THE DEEP REGIME

The aforementioned analysis has exposed the problems of the shallow regime: 1) shallow methods as base estimators have low potential in themselves; 2) there is a training gap between the primary task and subtasks, i.e., no representation learning is involved. Therefore, it is necessary to introduce our proposed $\mathcal{S}$-LDL of the deep regime, the overview of which is illustrated in Fig. 3. We illustrate our framework by introducing the learnable parts one by one.

Table 2: Modifications of different task adaptations

| Type | Subtask construction | $\ell_{\text{PRI}}$ | $\ell_{\text{SUB}}$ |
|---|---|---|---|
| Vanilla LDL | $(\boldsymbol{D}^{(1)}, \cdots) \leftarrow \text{SC}(\boldsymbol{D})$ | $\ell(\boldsymbol{D}, \tilde{\boldsymbol{D}}) \in \mathcal{L}_{\text{LDL}}$ | $\ell_{\text{SUB}}(\boldsymbol{D}^{(1)}, \cdots; \tilde{\boldsymbol{D}}^{(1)}, \cdots)$ |
| LDL4C | $(\boldsymbol{D}^{(1)}, \cdots) \leftarrow \text{SC}(\boldsymbol{D})$ | $\ell(\boldsymbol{D}, \bar{\boldsymbol{D}}, \tilde{\boldsymbol{D}}) \in \mathcal{L}_{\text{LDL4C}}$ | $\ell_{\text{SUB}}(\boldsymbol{D}^{(1)}, \cdots; \tilde{\boldsymbol{D}}^{(1)}, \cdots)$ |
| IncomLDL | $(\boldsymbol{D}_{\Omega}^{(1)}, \cdots) \leftarrow \text{SC}(\mathcal{R}_{\Omega}(\boldsymbol{D}))$ | $\ell(\mathcal{R}_{\Omega}(\boldsymbol{D}), \mathcal{R}_{\Omega}(\tilde{\boldsymbol{D}})) \in \mathcal{L}_{\text{IncomLDL}}$ | $\ell_{\text{SUB}}(\boldsymbol{D}_{\Omega}^{(1)}, \cdots; \mathcal{R}_{\Omega}(\tilde{\boldsymbol{D}}^{(1)}), \cdots)$ |
| LE | $(\boldsymbol{L}^{(1)}, \cdots) \leftarrow \text{SC}(\boldsymbol{L})$ | $\ell(\boldsymbol{L}, \tilde{\boldsymbol{D}}) \in \mathcal{L}_{\text{LE}}$ | $\ell_{\text{SUB}}(\boldsymbol{L}^{(1)}, \cdots; \tilde{\boldsymbol{D}}^{(1)}, \cdots)$ |

- $\varphi(\cdot)$ is guided by subtasks to learn a powerful representation, i.e., $\boldsymbol{R} = \varphi(\boldsymbol{X})$.

- $\psi(\cdot)$ is responsible for predicting subtask label distributions, i.e., $(\tilde{\boldsymbol{D}}^{(1)}, \cdots) = \psi(\boldsymbol{R})$. To ensure the precise prediction of subtask label distributions for reconstruction, we employ the mean absolute error function for subtask learning. The loss is weighted by the summation of the description degrees corresponding to the primary tasks, allowing more reliable label spaces to receive more attention. The subtask learning loss has the following form:

$$\ell_{\text{SUB}}\left(\mathcal{D}^{\circ}; \tilde{\mathcal{D}}^{\circ}\right) = \frac{1}{N|\mathcal{Y}^{\circ}|} \sum_{\mathcal{Y}^{(t)} \in \mathcal{Y}^{\circ}} \sum_{i=1}^{N} \left( \sum_{y_k \in \mathcal{Y}^{(t)}} d_{\boldsymbol{x}_i}^{y_k} \right) \sum_{j=1}^{|\mathcal{Y}^{(t)}|} \left| d_{\boldsymbol{x}_i}^{(t)y_j} - \tilde{d}_{\boldsymbol{x}_i}^{(t)y_j} \right|. \tag{7}$$

- $\omega(\cdot)$ can be any existing method that can be expressed as a network structure theoretically. Since the concatenation of the representation and subtask label distributions, we have $\boldsymbol{Z} = (\boldsymbol{R}, \psi(\boldsymbol{R}))$ and $\tilde{\boldsymbol{D}} = \omega(\boldsymbol{Z})$. In the case of the primary task being vanilla LDL, the primary task loss $\ell_{\text{PRI}}$ can be

$$\ell_{\text{KL}}\left(\boldsymbol{D}, \tilde{\boldsymbol{D}}\right) = \frac{1}{N} \sum_{i=1}^{N} \sum_{j=1}^{L} d_{\boldsymbol{x}_i}^{y_j} \ln \frac{d_{\boldsymbol{x}_i}^{y_j}}{\tilde{d}_{\boldsymbol{x}_i}^{y_j}}, \quad \ell_{\text{KL}} \in \mathcal{L}_{\text{LDL}}. \tag{8}$$

Note that $\ell_{\text{PRI}}$ changes as the primary task changes. Finally, we can learn the model parameters $\boldsymbol{\Theta}$ by

$$\boldsymbol{\Theta}^* = \arg\min_{\boldsymbol{\Theta}}(\ell_{\text{PRI}} + \alpha \ell_{\text{SUB}}), \tag{9}$$

where $\alpha$ is a trade-off parameter. Compared with the shallow regime, $\mathcal{S}$-LDL of the deep regime has the following advantages: 1) There is no two-stage training gap, which makes the representation contain insights from both the primary task and the subtasks; 2) the framework not only serves LDL, but can also be directly applied to derivative tasks of LDL, e.g., LDL for classification (LDL4C) (Wang, Jing and Geng, Xin, 2019), incomplete LDL (IncomLDL) (Xu and Zhou, 2017), label enhancement (LE) (Xu et al., 2019). The modifications involved are shown in Table 2, where $\mathcal{L}_{\text{X}}$ indicates the set of losses for adaptable methods in the task of type "X". Special mathematical procedures of LDL4C and IncomLDL are defined as

$$\left[\bar{\boldsymbol{D}}^{(t)}\right]_{ij} \triangleq \begin{cases} 1, & \text{if } y_j = \arg\max_{\bar{y} \in \mathcal{Y}^{(t)}} d_{\boldsymbol{x}_i}^{\bar{y}} \\ 0, & \text{otherwise} \end{cases}, \quad \left[\mathcal{R}_{\Omega}(\boldsymbol{D})\right]_{ij} \triangleq \begin{cases} [\boldsymbol{D}]_{ij}, & \text{if } (i, y_j) \in \Omega \\ 0, & \text{otherwise} \end{cases}, \tag{10}$$

respectively, where $[\cdot]_{ij}$ represents the element in $i$-th row of the matrix corresponding to label $y_j$, and $\Omega$ represents observed elements sampled uniformly at random from $\boldsymbol{D}$ in IncomLDL. Such modifications are rational since: 1) targets of LDL4C and IncomLDL, i.e., $\bar{\boldsymbol{D}}$ and $\mathcal{R}_{\Omega}(\boldsymbol{D})$, are essentially different forms of degradation of the label distribution matrix; and 2) the target of LE is a logical label matrix $\boldsymbol{L}$, the same as the target of MLL, which is actually a special case of LDL.

## 6 EXPERIMENTS

In this section, we evaluate $\mathcal{S}$-LDL of the deep regime. Due to page limitations, datasets, comparison methods, and their parameter settings are introduced in the appendix.

**Metrics** For LDL, we use the same metrics suggested by Jia et al. (2023). Due to page limitations, we only present results on Cheby. ↓ (Chebyshev distance), Clark ↓ (Clark distance), Cosine ↑ (cosine similarity), and Spear. ↑ (Spearman's coefficient) in the main paper, where ↓ (↑) indicates "the lower (higher) the better". Note that these metrics are *not* as intuitive as accuracy or error rate, i.e., *small changes can mean large performance differences.* For LDL4C, objective of which is different from LDL, we use 0/1 loss ↓ (zero one loss) and Err. prob. ↓ (error probability) as metrics (Wang, Jing and Geng, Xin, 2019).

Table 3: Experimental results of LDL on `JAFFE` and `Yeast_diau` formatted as $(\text{mean} \pm \text{std}(\text{rank}))$

| Algorithms | JAFFE (Lyons et al., 1998) | | Algorithms | Yeast_diau (Geng, 2016) | |
| --- | --- | --- | --- | --- | --- |
| | Clark ↓ | Cosine ↑ | | Cheby. ↓ | Spear. ↑ |
| LDSVR (Geng and Hou, 2015) | $.3280 \pm .027$ (6) | $.9549 \pm .010$ (7) | CPNN (Geng et al., 2013) | $.0385 \pm .001$ (9) | $.2962 \pm .034$ (10) |
| AA-$k$NN (Geng, 2016) | $.3483 \pm .032$ (8) | $.9497 \pm .010$ (9) | AA-$k$NN | $.0385 \pm .001$ (9) | $.3674 \pm .029$ (9) |
| LDLFs (Shen et al., 2017) | $.3637 \pm .032$ (10) | $.9494 \pm .009$ (10) | LDLFs | $.0371 \pm .001$ (8) | $.4088 \pm .021$ (8) |
| DF-BFGS (González et al., 2021a) | $.3062 \pm .025$ (3) | $.9633 \pm .007$ (2) | DF-BFGS | $.0368 \pm .001$ (5) | $.4161 \pm .027$ (5) |
| KLD (Geng, 2016) ● | $.3608 \pm .031$ (9) | $.9538 \pm .008$ (8) | LRR ● | $.0370 \pm .001$ (7) | $.4154 \pm .023$ (6) |
| $\mathcal{S}$-KLD | $.3007 \pm .032$ (2) | $.9625 \pm .009$ (3) | $\mathcal{S}$-LRR | $\mathbf{.0366} \pm .001$ (1) | $.4198 \pm .023$ (2) |
| SCL (Jia et al., 2019) ● | $.3358 \pm .024$ (7) | $.9592 \pm .006$ (6) | QFD$^2$ (Wen et al., 2023) ● | $.0369 \pm .001$ (6) | $.4118 \pm .025$ (7) |
| $\mathcal{S}$-SCL | $.3184 \pm .025$ (4) | $.9604 \pm .008$ (5) | $\mathcal{S}$-QFD$^2$ | $\mathbf{.0366} \pm .001$ (1) | $\mathbf{.4203} \pm .021$ (1) |
| LRR (Jia et al., 2023) ● | $.3230 \pm .027$ (5) | $.9616 \pm .008$ (4) | CJS (Wen et al., 2023) | $.0367 \pm .001$ (4) | $.4164 \pm .025$ (4) |
| $\mathcal{S}$-LRR | $\mathbf{.2934} \pm .028$ (1) | $\mathbf{.9635} \pm .008$ (1) | $\mathcal{S}$-CJS | $\mathbf{.0366} \pm .001$ (1) | $.4198 \pm .024$ (2) |

Table 4: Experimental results of LDL4C on `sBU_3DFE` and `Flickr` formatted as $(\text{mean} \pm \text{std}(\text{rank}))$

| Algorithms | sBU_3DFE (Geng, 2016) | | Algorithms | Flickr (Yang et al., 2017b) | |
| --- | --- | --- | --- | --- | --- |
| | 0/1 loss ↓ | Err. prob. ↓ | | 0/1 loss ↓ | Err. prob. ↓ |
| LDL4C (Wang, Jing and Geng, Xin, 2019) | $.5578 \pm .028$ (6) | $.7671 \pm .007$ (5) | LDL4C | $.8971 \pm .008$ (6) | $.8884 \pm .004$ (6) |
| $\mathcal{S}$-LDL4C | $.5526 \pm .025$ (5) | $.7686 \pm .006$ (6) | $\mathcal{S}$-LDL4C | $.8705 \pm .138$ (5) | $.8702 \pm .100$ (5) |
| HR (Wang and Geng, 2021a) ● | $.5167 \pm .027$ (3) | $.7596 \pm .006$ (2) | HR ● | $.4513 \pm .015$ (4) | $.5823 \pm .007$ (4) |
| $\mathcal{S}$-HR | $.5069 \pm .025$ (2) | $.7598 \pm .006$ (3) | $\mathcal{S}$-HR | $\mathbf{.4219} \pm .015$ (1) | $\mathbf{.5639} \pm .007$ (1) |
| LDLM (Wang and Geng, 2021b) ● | $.5258 \pm .034$ (4) | $.7619 \pm .009$ (4) | LDLM ● | $.4384 \pm .014$ (3) | $.5740 \pm .007$ (3) |
| $\mathcal{S}$-LDLM | $\mathbf{.4809} \pm .024$ (1) | $\mathbf{.7524} \pm .005$ (1) | $\mathcal{S}$-LDLM | $.4321 \pm .016$ (2) | $.5667 \pm .007$ (2) |

**Results and discussion** We apply $\mathcal{S}$-LDL to existing methods to demonstrate performance improvements. For each dataset we conduct ten-fold experiments repeated 10 times, and the average performance is recorded. Tables 3 to 4 show representative results and the remainder are in the appendix, where ● (○) indicates that more than half of the metrics support that "$\mathcal{S}$-X" is statistically superior (inferior) to the corresponding methods "X" (pairwise $t$-test at 0.05 significance level); there is no significant if neither ● nor ○ is shown. LRR focuses on the label ranking relationship, which is also emphasized by each subtask. We believe this is why $\mathcal{S}$-LDL and LRR fit so well. Note that our method has the least improvement in SCL, which may be

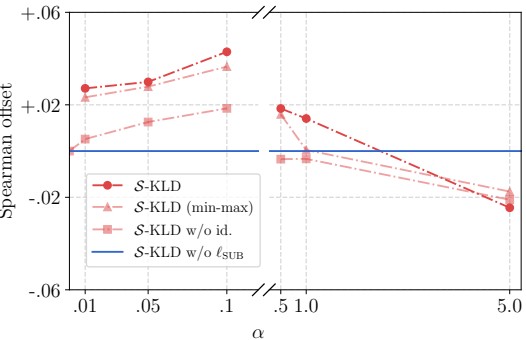

Figure 4: Visualized results of the ablation study and the parameter sensitivity analysis, which is on the `Natural_Scene` (Geng, 2016) dataset.

attributed to its reliance on shallow regime methods in the prediction phase. It is also worth noting that the improvement in vanilla KLD is considerable, which just illustrates the limitations of loss function engineering that considers label correlation one-sidedly. QFD$^2$ and CJS are tailored for ordinary LDL, and may have better results than LRR on this regard. Powered by $\mathcal{S}$-LDL, these methods can all achieve better level. For LDL4C, $\mathcal{S}$-LDL significantly improves both HR and LDLM. However, it can be observed that $\mathcal{S}$-LDL4C is unstable on the `Flickr` dataset, which is not surprising since LDL4C itself fails on it. We believe this is caused by the combined effect of the sparsity of the dataset and the information entropy operation involved in LDL4C.

**Parameter sensitivity** We check the sensitivity of the trade-off parameter $\alpha$ on the LDL task with the `Natural_Scene` dataset by varying the parameter in $\{0.01, 0.05, 0.1, 0.5, 1, 5\}$. Results are shown in Fig. 4. Spearman's coefficient of $\mathcal{S}$-LDL first increases and then decreases as $\alpha$ varies, demonstrating a desirable bell-shaped curve. This justifies our motivation of jointly learning the primary task and subtasks, as a good trade-off between them can enhance the performance.

**Ablation study** Here we are interested in the importance of each part of $\mathcal{S}$-LDL, thus an ablation study is performed with $\mathcal{S}$-KLD: 1) we replace $\mathcal{N}_{\text{SUM}}$ in SC with the min-max function to examine the importance of the subtask distribution reconstruction, and this model is denoted as $\mathcal{S}$-KLD (min-max); 2) we remove the identity mapping in Fig. 3 to examine the importance of the prediction via subtask representation, and this model is denoted as $\mathcal{S}$-KLD w/o id.; 3) we train without the term of $\ell_{\text{SUB}}$ (i.e.,

setting $\alpha = 0$) to examine the importance of subtask learning, and this model is denoted as $\mathcal{S}$-KLD w/o $\ell_{\text{SUB}}$. Results are also shown in Fig. 4, which confirms that each part of $\mathcal{S}$-LDL contributes as long as there is a good trade-off.

## 7 LIMITATIONS AND CONCLUSION

**Limitations**   First, $\mathcal{S}$-LDL of the shallow regime is proposed out of intuition, and in Section 5, we have discussed its limitations, which are addressed via the designing of $\mathcal{S}$-LDL of the deep regime. Second, when the label space is large, especially when labels are continuous and result in unimodal label distributions (e.g., age estimation), our proposed cannot be rationally applied. Fortunately, one possible workaround is to use a binning tricks for preprocessing, and then construct subtasks.

**Conclusion**   We propose $\mathcal{S}$-LDL, a subtask learning framework nested into LDL. $\mathcal{S}$-LDL is generic: it generates pseudo-supervised information via subtask construction without any extra knowledge; $\mathcal{S}$-LDL is minimalist: it can be attached to existing methods and handle derivative tasks; $\mathcal{S}$-LDL is efficient: it captures a wide variety of label correlations. The analysis shows the validity and reconstructability of subtasks, and experiments show the superiority of our framework.

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

## A  APPENDIX: DETAILS OF EXPERIMENTS

Here we try our best to provide as much information as possible for reproducible research.

### A.1  METRICS

For LDL, IncomLDL, and LE, we use the same metrics suggested by Geng (2016), which are `Cheby.` ↓ (Chebyshev distance), `Clark` ↓ (Clark distance), `Can.` ↓ (Canberra distance), `KLD` ↓ (Kullback-Leibler divergence), `Cosine` ↑ (cosine similarity), and `Int.` ↑ (intersection similarity), respectively. Here ↓ (↑) indicates "the lower (higher) the better". For LDL and IncomLDL, we additionally use two ranking metrics: `Spear.` ↑ (Spearman's coefficient) and `Ken.` ↑ (Kendall's coefficient) (Jia et al., 2023). Note that these metrics are *not* as intuitive as accuracy or error rate, i.e., *small changes can mean large performance differences.* For LDL4C, objective of which is different from LDL, we use $0/1$ `loss` ↓ (zero one loss) and `Err. prob.` ↓ (error probability) as metrics (Wang, Jing and Geng, Xin, 2019). Let the real distribution be denoted by $\boldsymbol{u} = \{u_j\}_{j=1}^{L}$, and the predicted distribution be denoted by $\boldsymbol{v} = \{v_j\}_{j=1}^{L}$, then the above metrics can be summarized in Table 5, where $\rho(\cdot)$ and $\delta(\cdot, \cdot)$ are the ranking function and the Kronecker delta function, respectively.

Table 5: Summary of the metrics

| Name | Formula | Name | Formula |
|---|---|---|---|
| Cheby. $\downarrow$ | $\text{Dis}_1(\boldsymbol{u},\,\boldsymbol{v}) = \max_j |u_j - v_j|$ | Cosine $\uparrow$ | $\text{Sim}_1(\boldsymbol{u},\,\boldsymbol{v}) = \frac{\sum_{j=1}^L u_j v_j}{\sqrt{\sum_{j=1}^L u_j^2}\sqrt{\sum_{j=1}^L v_j^2}}$ |
| Clark $\downarrow$ | $\text{Dis}_2(\boldsymbol{u},\,\boldsymbol{v}) = \sqrt{\sum_{j=1}^L \frac{(u_j - v_j)^2}{(u_j + v_j)^2}}$ | Int. $\uparrow$ | $\text{Sim}_2(\boldsymbol{u},\,\boldsymbol{v}) = \sum_{j=1}^L \min(u_j,\, v_j)$ |
| Can. $\downarrow$ | $\text{Dis}_3(\boldsymbol{u},\,\boldsymbol{v}) = \sum_{j=1}^L \frac{|u_j - v_j|}{u_j + v_j}$ | Spear. $\uparrow$ | $\text{Rnk}_1(\boldsymbol{u},\,\boldsymbol{v}) = 1 - \frac{6\sum_{j=1}^L (\rho(u_j) - \rho(v_j))^2}{L(L^2 - 1)}$ |
| KLD $\downarrow$ | $\text{Dis}_4(\boldsymbol{u},\,\boldsymbol{v}) = \sum_{j=1}^L u_j \ln \frac{u_j}{v_j}$ | Ken. $\uparrow$ | $\text{Rnk}_2(\boldsymbol{u},\,\boldsymbol{v}) = \frac{2\sum_{j<k} \text{sgn}(u_j - u_k)\text{sgn}(v_j - v_k)}{L(L-1)}$ |
| 0/1 loss $\downarrow$ | $C_1(\boldsymbol{u},\,\boldsymbol{v}) = \delta(\arg\max(\boldsymbol{u}),\ \arg\max(\boldsymbol{v}))$ | Err. prob. $\downarrow$ | $C_2(\boldsymbol{u},\,\boldsymbol{v}) = 1 - u_{\arg\max(\boldsymbol{v})}$ |

## A.2 DATASETS

We adopt several widely used label distribution datasets, including: JAFFE (Lyons et al., 1998);[4] fbp5500 (Liang et al., 2018);[5] sBU_3DFE, Movie, Natural_Scene, Yeast_heat, Yeast_diau, Yeast_cold, and Yeast_dtt provided by Geng (2016);[6] emotion6, Twitter, and Flickr provided by Yang et al. (2017b).[7] The information of these datasets are summarized in Table 6.

## A.3 COMPARISON METHODS

On the one hand, we apply our proposed $\mathcal{S}$-LDL to existing methods to demonstrate performance improvements in the LDL task (denoted by the "$\mathcal{S}$-" prefix). These methods are BFGS-LLD (KLD) (Geng, 2016), SCL (Jia et al., 2019), LRR (Jia et al., 2023), QFD$^2$ (Wen et al., 2023), and CJS (Wen et al., 2023) (the losses of these methods constitute the set $\mathcal{L}_{\text{LDL}}$). On the other hand, we compare $\mathcal{S}$-LDL with methods that have specialized structure, which our proposed cannot directly adapt to. These methods are

Table 6: Summary of datasets

| Dataset | # Instances $N$ | # Features $P$ | # Labels $L$ |
|---|---|---|---|
| JAFFE | 213 | 243 | 6 |
| sBU_3DFE | 2500 | 243 | 6 |
| Movie | 7755 | 1869 | 5 |
| Nature_Scene | 2000 | 294 | 9 |
| fbp5500 | 5500 | 512 | 5 |
| Yeast_heat | 2465 | 24 | 6 |
| Yeast_diau | 2465 | 24 | 7 |
| Yeast_cold | 2465 | 24 | 4 |
| Yeast_dtt | 2465 | 24 | 4 |
| emotion6 | 1980 | 168 | 7 |
| Twitter | 10045 | 168 | 8 |
| Flickr | 11150 | 168 | 8 |

CPNN (Geng et al., 2013), LDSVR (Geng and Hou, 2015), AA-$k$NN (Geng, 2016), LDLFs (Shen et al., 2017), and DF-LDL (denoted by DF-BFGS since we use BFGS-LLDs as base estimators) (González et al., 2021a). Moreover, we apply our proposed to derivative tasks of LDL (i.e., LDL4C, IncomLDL, and LE) and the comparison methods involved are LDL4C (Wang, Jing and Geng, Xin, 2019), HR (Wang and Geng, 2021a), LDLM (Wang and Geng, 2021b), IncomLDL (Xu and Zhou, 2017), LP (Xu et al., 2019), GLLE (Xu et al., 2019), LEVI (Xu et al., 2023), and LIBLE (Zheng et al., 2023).

## A.4 PARAMETER SETTINGS AND EXPERIMENTAL ENVIRONMENT

The parameter settings of the proposed $\mathcal{S}$-LDL and comparison algorithms are summarized in Table 7. Note that DF-LDL is parameter-free, and we use BFGS-LLDs as its base estimators, parameter settings of which are the same as BFGS-LLD as the comparison algorithm. We use Adam (Kingma and Ba, 2015) for the optimization of $\mathcal{S}$-LDL. For all methods of the deep regime, the learning rate is chosen among $\{1,\, 2,\, 5\} \times 10^{\{-4,\, -3,\, -2\}}$, and the selection of the number of epochs is nested into

---

[4] https://zenodo.org/records/3451524

[5] https://github.com/HCIILAB/SCUT-FBP5500-Database-Release

[6] https://palm.seu.edu.cn/xgeng/LDL/download.htm

[7] https://cv.nankai.edu.cn/projects

Table 7: Summary of algorithms and parameter settings

| Algorithms | Parameter | Value (Range) |
|---|---|---|
| AA-$k$NN | $k$: # Neighbors | 5 |
| LDLFs | # Estimators (trees) | 5 |
| | Depth | 6 |
| | Latent units (leaves) | 64 |
| BFGS-LLD | $\varepsilon$: Convergence criterion | $10^{-6}$ |
| | Max iteration | 600 |
| SCL | $m$: # Clusters | 5 |
| | $\lambda_1, \lambda_2, \lambda_3$: Trade-off | $10^{-3}, 10^{-3}, 0.1$ |
| LRR | $\lambda$: Trade-off (ranking loss) | $10^{\{-5,-4,-3,-2,-1\}}$ |
| | $\beta$: Trade-off (regularization) | $10^{\{-3,-2,-1,0,1,2\}}$ |
| LDL4C | $C_1, C_2$: Balance coefficients | $10^{-2}, 10^{-6}$ |
| | $\rho$: Margin | $10^{-2}$ |
| HR | $\lambda_1, \lambda_2, \lambda_3$: Trade-off | $10^{-2}, 10^{-6}$ |
| | $\rho$: Margin | $10^{-2}$ |
| LDLM | $\lambda_1, \lambda_2, \lambda_3$: Trade-off | $10^{-6}, 10^{\{-3,-2,-1\}}, 10^{\{-3,-2,-1\}}$ |
| | $\rho$: Margin | $10^{-2}$ |
| IncomLDL | $\varepsilon$: Convergence criterion | $10^{-6}$ |
| | $\gamma$: Factor of Lipschitz constant | 2 |
| | $\lambda$: Trade-off | 1 |
| LP | $\alpha$: Balance coefficient | 0.5 |
| GLLE | $\lambda_1, \lambda_2$: Trade-off | $10^{-2}, 10^{-4}$ |
| | $\sigma$: Width parameter for similarity calculation | 10 |
| LEVI | $\lambda$: Trade-off | 1 |
| LIBLE | $\alpha, \beta$: Trade-off | $10^{\{-3,-2,-1,0,1,2\}}$ |
| $\mathcal{S}$-LDL | $\alpha, \lambda, T$ | 0.1, 0.2, 10 |

a ten-fold cross validation. All the results are obtained on a Linux workstation with Intel Core i9 (3.70GHz), NVIDIA GeForce RTX 3090 (24GB), and 32GB memory.

## A.5 FULL EXPERIMENTAL RESULTS

Here we provide complete results of all conducted experiments. Tables 8 to 19 are results on the LDL task with different datasets. For IncomLDL, we follow *the incomplete settings* (Xu and Zhou, 2017) and vary the observed rate $\omega\%$ from 20% to 40%. Tables 20 to 21 are results on the IncomLDL task. Tables 22 to 23 are on the LDL4C task. For LE, we follow *the settings of the recovery experiment* (Xu et al., 2019). Tables 24 to 25 show results on the LE task.

Table 8: Experimental results of LDL on the `JAFFE` dataset formatted as (`mean` $\pm$ `std`)

| Algorithms | Cheby. ↓ | Clark ↓ | Can. ↓ | KLD ↓ | Cosine ↑ | Int. ↑ | Spear. ↑ | Ken. ↑ |
|---|---|---|---|---|---|---|---|---|
| LDSVR | $.0959_{\pm.013}$ | $.3280_{\pm.027}$ | $.6778_{\pm.058}$ | $.0476_{\pm.011}$ | $.9549_{\pm.010}$ | $.8838_{\pm.012}$ | $.5175_{\pm.102}$ | $.4508_{\pm.086}$ |
| AA-$k$NN | $.0978_{\pm.012}$ | $.3483_{\pm.032}$ | $.7164_{\pm.066}$ | $.0527_{\pm.011}$ | $.9497_{\pm.010}$ | $.8766_{\pm.012}$ | $.4111_{\pm.083}$ | $.3514_{\pm.070}$ |
| LDLFs | $.0940_{\pm.010}$ | $.3637_{\pm.032}$ | $.7355_{\pm.066}$ | $.0550_{\pm.009}$ | $.9494_{\pm.009}$ | $.8766_{\pm.011}$ | $.4364_{\pm.108}$ | $.3749_{\pm.093}$ |
| DF-BFGS | $.0827_{\pm.009}$ | $.3062_{\pm.025}$ | $.6239_{\pm.052}$ | $.0388_{\pm.007}$ | $.9633_{\pm.007}$ | $.8944_{\pm.010}$ | $.5244_{\pm.087}$ | $.4493_{\pm.077}$ |
| KLD ● | $.0925_{\pm.010}$ | $.3608_{\pm.031}$ | $.7363_{\pm.064}$ | $.0508_{\pm.009}$ | $.9538_{\pm.008}$ | $.8777_{\pm.011}$ | $.4572_{\pm.097}$ | $.3873_{\pm.084}$ |
| $\mathcal{S}$-KLD | $.0818_{\pm.011}$ | $.3007_{\pm.032}$ | $.6132_{\pm.067}$ | $.0395_{\pm.010}$ | $.9625_{\pm.009}$ | $.8960_{\pm.012}$ | $\mathbf{.5461}_{\pm.105}$ | $.4769_{\pm.096}$ |
| SCL ● | $.0873_{\pm.008}$ | $.3358_{\pm.024}$ | $.6874_{\pm.051}$ | $.0439_{\pm.006}$ | $.9592_{\pm.006}$ | $.8851_{\pm.009}$ | $.4744_{\pm.092}$ | $.4020_{\pm.080}$ |
| $\mathcal{S}$-SCL | $.0854_{\pm.010}$ | $.3184_{\pm.025}$ | $.6526_{\pm.053}$ | $.0420_{\pm.008}$ | $.9604_{\pm.008}$ | $.8896_{\pm.010}$ | $.5110_{\pm.095}$ | $.4388_{\pm.084}$ |
| LRR ● | $.0853_{\pm.010}$ | $.3230_{\pm.027}$ | $.6560_{\pm.055}$ | $.0412_{\pm.008}$ | $.9616_{\pm.008}$ | $.8906_{\pm.010}$ | $.5117_{\pm.094}$ | $.4420_{\pm.084}$ |
| $\mathcal{S}$-LRR | $\mathbf{.0804}_{\pm.009}$ | $\mathbf{.2934}_{\pm.028}$ | $\mathbf{.5989}_{\pm.059}$ | $\mathbf{.0383}_{\pm.009}$ | $\mathbf{.9635}_{\pm.008}$ | $.8981_{\pm.011}$ | $.5448_{\pm.092}$ | $\mathbf{.4819}_{\pm.084}$ |

Table 9: Experimental results of LDL on the sBU_3DFE dataset formatted as (mean ± std)

| Algorithms | Cheby. ↓ | Clark ↓ | Can. ↓ | KLD ↓ | Cosine ↑ | Int. ↑ | Spear. ↑ | Ken. ↑ |
|---|---|---|---|---|---|---|---|---|
| LDSVR | .1250 ±.005 | .3710 ±.010 | .8009 ±.021 | .0720 ±.004 | .9298 ±.004 | .8559 ±.004 | .3524 ±.031 | .3011 ±.026 |
| AA-$k$NN | .1272 ±.004 | .4001 ±.009 | .8281 ±.020 | .0801 ±.004 | .9217 ±.004 | .8488 ±.004 | .2053 ±.030 | .1767 ±.026 |
| LDLFs | .1016 ±.003 | .3262 ±.008 | .6841 ±.017 | .0504 ±.003 | .9499 ±.003 | .8776 ±.003 | .4212 ±.023 | .3620 ±.019 |
| DF-BFGS | .1146 ±.004 | .3616 ±.008 | .7627 ±.019 | .0618 ±.003 | .9388 ±.003 | .8626 ±.004 | .3026 ±.031 | .2621 ±.026 |
| KLD ● | .1147 ±.004 | .3697 ±.008 | .7804 ±.019 | .0624 ±.003 | .9387 ±.003 | .8604 ±.003 | .3021 ±.026 | .2643 ±.022 |
| $\mathcal{S}$-KLD | .1014 ±.004 | .3203 ±.009 | .6736 ±.018 | .0514 ±.003 | .9487 ±.003 | .8789 ±.004 | .4334 ±.025 | .3729 ±.022 |
| SCL ● | .1145 ±.004 | .3648 ±.008 | .7748 ±.018 | .0605 ±.003 | .9404 ±.003 | .8614 ±.003 | .3091 ±.026 | .2701 ±.021 |
| $\mathcal{S}$-SCL | .1041 ±.004 | .3301 ±.009 | .6936 ±.019 | .0535 ±.003 | .9468 ±.003 | .8754 ±.004 | .3956 ±.030 | .3381 ±.027 |
| LRR ● | .1067 ±.003 | .3476 ±.008 | .7320 ±.017 | .0543 ±.003 | .9465 ±.003 | .8695 ±.003 | .3626 ±.026 | .3123 ±.022 |
| $\mathcal{S}$-LRR | **.0996** ±.004 | **.3157** ±.008 | **.6610** ±.017 | **.0499** ±.003 | **.9502** ±.003 | **.8812** ±.003 | **.4455** ±.026 | **.3837** ±.023 |

Table 10: Experimental results of LDL on the Yeast_heat dataset formatted as (mean ± std)

| Algorithms | Cheby. ↓ | Clark ↓ | Can. ↓ | KLD ↓ | Cosine ↑ | Int. ↑ | Spear. ↑ | Ken. ↑ |
|---|---|---|---|---|---|---|---|---|
| CPNN | .0419 ±.001 | .1818 ±.005 | .3633 ±.009 | .0125 ±.001 | .9881 ±.001 | .9404 ±.001 | .1507 ±.034 | .1221 ±.028 |
| AA-$k$NN | .0441 ±.001 | .1913 ±.005 | .3840 ±.010 | .0140 ±.001 | .9867 ±.001 | .9370 ±.002 | .1678 ±.031 | .1384 ±.026 |
| LDLFs | .0420 ±.001 | .1818 ±.005 | .3627 ±.009 | .0125 ±.001 | .9881 ±.001 | .9405 ±.001 | .1731 ±.032 | .1409 ±.026 |
| DF-BFGS | .0420 ±.001 | .1816 ±.005 | .3624 ±.009 | .0125 ±.001 | .9881 ±.001 | .9405 ±.001 | **.1964** ±.034 | **.1624** ±.028 |
| LRR ● | .0423 ±.001 | .1828 ±.005 | .3644 ±.009 | .0126 ±.001 | .9880 ±.001 | .9402 ±.001 | .1655 ±.033 | .1351 ±.028 |
| $\mathcal{S}$-LRR | **.0417** ±.001 | .1806 ±.005 | .3609 ±.009 | **.0124** ±.001 | **.9882** ±.001 | .9408 ±.001 | .1882 ±.034 | .1548 ±.028 |
| QFD$^2$ ● | .0423 ±.001 | .1827 ±.005 | .3644 ±.009 | .0126 ±.001 | .9880 ±.001 | .9402 ±.001 | .1677 ±.032 | .1351 ±.027 |
| $\mathcal{S}$-QFD$^2$ | **.0417** ±.001 | .1808 ±.005 | .3611 ±.009 | **.0124** ±.001 | **.9882** ±.001 | .9408 ±.001 | .1880 ±.032 | .1544 ±.027 |
| CJS ● | .0423 ±.001 | .1827 ±.005 | .3643 ±.009 | .0126 ±.001 | .9880 ±.001 | .9402 ±.001 | .1632 ±.032 | .1329 ±.027 |
| $\mathcal{S}$-CJS | **.0417** ±.001 | **.1804** ±.005 | **.3603** ±.009 | **.0124** ±.001 | **.9882** ±.001 | **.9409** ±.001 | .1940 ±.030 | .1589 ±.025 |

Table 11: Experimental results of LDL on the Yeast_diau dataset formatted as (mean ± std)

| Algorithms | Cheby. ↓ | Clark ↓ | Can. ↓ | KLD ↓ | Cosine ↑ | Int. ↑ | Spear. ↑ | Ken. ↑ |
|---|---|---|---|---|---|---|---|---|
| CPNN | .0385 ±.001 | .2069 ±.006 | .4439 ±.012 | .0138 ±.001 | .9872 ±.001 | .9383 ±.002 | .2962 ±.034 | .2427 ±.027 |
| AA-$k$NN | .0385 ±.001 | .2085 ±.006 | .4487 ±.014 | .0145 ±.001 | .9867 ±.001 | .9377 ±.002 | .3674 ±.029 | .2976 ±.024 |
| LDLFs | .0371 ±.001 | .2014 ±.006 | .4324 ±.012 | .0132 ±.001 | .9879 ±.001 | .9401 ±.002 | .4088 ±.021 | .3254 ±.018 |
| DF-BFGS | .0368 ±.001 | .1999 ±.006 | .4294 ±.013 | .0131 ±.001 | .9879 ±.001 | .9405 ±.002 | .4161 ±.027 | **.3404** ±.022 |
| LRR ● | .0370 ±.001 | .2007 ±.006 | .4307 ±.012 | .0131 ±.001 | .9879 ±.001 | .9403 ±.002 | .4154 ±.023 | .3343 ±.020 |
| $\mathcal{S}$-LRR | **.0366** ±.001 | **.1983** ±.006 | **.4257** ±.012 | **.0129** ±.001 | **.9881** ±.001 | **.9410** ±.002 | .4198 ±.023 | .3389 ±.019 |
| QFD$^2$ ● | .0369 ±.001 | .2000 ±.006 | .4296 ±.012 | .0131 ±.001 | .9879 ±.001 | .9404 ±.002 | .4118 ±.025 | .3326 ±.021 |
| $\mathcal{S}$-QFD$^2$ | **.0366** ±.001 | .1985 ±.006 | .4261 ±.012 | **.0129** ±.001 | **.9881** ±.001 | .9409 ±.002 | **.4203** ±.021 | .3387 ±.018 |
| CJS | .0367 ±.001 | .1989 ±.006 | .4272 ±.012 | .0130 ±.001 | .9880 ±.001 | .9408 ±.002 | .4164 ±.025 | .3366 ±.021 |
| $\mathcal{S}$-CJS | **.0366** ±.001 | .1984 ±.006 | .4260 ±.012 | .0130 ±.001 | **.9881** ±.001 | .9409 ±.002 | .4198 ±.024 | .3392 ±.019 |

Table 12: Experimental results of LDL on the Yeast_cold dataset formatted as (mean ± std)

| Algorithms | Cheby. ↓ | Clark ↓ | Can. ↓ | KLD ↓ | Cosine ↑ | Int. ↑ | Spear. ↑ | Ken. ↑ |
|---|---|---|---|---|---|---|---|---|
| CPNN | **.0510** ±.002 | .1392 ±.005 | .2396 ±.008 | **.0121** ±.001 | **.9886** ±.001 | **.9410** ±.002 | **.2651** ±.036 | **.2263** ±.032 |
| AA-$k$NN | .0542 ±.002 | .1476 ±.005 | .2549 ±.008 | .0135 ±.001 | .9872 ±.001 | .9371 ±.002 | .2189 ±.035 | .1866 ±.031 |
| LDLFs | .0511 ±.002 | .1396 ±.005 | .2404 ±.009 | .0122 ±.001 | .9885 ±.001 | .9408 ±.002 | .2482 ±.038 | .2112 ±.033 |
| DF-BFGS | .0514 ±.002 | .1404 ±.005 | .2424 ±.008 | .0123 ±.001 | .9885 ±.001 | .9403 ±.002 | .2581 ±.036 | .2190 ±.030 |
| LRR | .0511 ±.002 | .1395 ±.005 | .2402 ±.009 | .0122 ±.001 | **.9886** ±.001 | .9408 ±.002 | .2490 ±.035 | .2111 ±.030 |
| $\mathcal{S}$-LRR | **.0510** ±.002 | **.1391** ±.005 | **.2395** ±.009 | **.0121** ±.001 | **.9886** ±.001 | **.9410** ±.002 | .2618 ±.037 | .2238 ±.032 |
| QFD$^2$ | .0513 ±.002 | .1401 ±.005 | .2413 ±.009 | .0123 ±.001 | .9885 ±.001 | .9405 ±.002 | .2534 ±.037 | .2158 ±.032 |
| $\mathcal{S}$-QFD$^2$ | **.0510** ±.002 | **.1391** ±.005 | .2396 ±.008 | **.0121** ±.001 | **.9886** ±.001 | **.9410** ±.002 | .2571 ±.039 | .2197 ±.033 |
| CJS | .0513 ±.002 | .1401 ±.005 | .2412 ±.008 | .0123 ±.001 | .9884 ±.001 | .9406 ±.002 | .2535 ±.038 | .2152 ±.032 |
| $\mathcal{S}$-CJS | **.0510** ±.002 | .1392 ±.005 | .2396 ±.009 | **.0121** ±.001 | **.9886** ±.001 | **.9410** ±.002 | .2621 ±.037 | .2241 ±.031 |

Table 13: Experimental results of LDL on the `Yeast_dtt` dataset formatted as $(\text{mean} \pm \text{std})$

| Algorithms | Cheby. ↓ | Clark ↓ | Can. ↓ | KLD ↓ | Cosine ↑ | Int. ↑ | Spear. ↑ | Ken. ↑ |
|---|---|---|---|---|---|---|---|---|
| CPNN | $.0361_{\pm.001}$ | $.0984_{\pm.004}$ | $.1690_{\pm.006}$ | $.0063_{\pm.001}$ | $.9941_{\pm.000}$ | $.9583_{\pm.001}$ | $.1735_{\pm.035}$ | $.1494_{\pm.030}$ |
| AA-$k$NN | $.0386_{\pm.001}$ | $.1047_{\pm.004}$ | $.1797_{\pm.006}$ | $.0071_{\pm.001}$ | $.9933_{\pm.000}$ | $.9556_{\pm.001}$ | $.1591_{\pm.033}$ | $.1399_{\pm.030}$ |
| LDLFs | $.0360_{\pm.001}$ | $.0981_{\pm.004}$ | $.1689_{\pm.006}$ | $.0063_{\pm.001}$ | $\mathbf{.9941}_{\pm.000}$ | $.9583_{\pm.001}$ | $.1986_{\pm.038}$ | $.1727_{\pm.034}$ |
| DF-BFGS | $.0365_{\pm.001}$ | $.0995_{\pm.004}$ | $.1712_{\pm.006}$ | $.0064_{\pm.001}$ | $.9939_{\pm.000}$ | $.9578_{\pm.001}$ | $.1804_{\pm.033}$ | $.1592_{\pm.030}$ |
| LRR | $.0360_{\pm.001}$ | $.0982_{\pm.004}$ | $.1690_{\pm.006}$ | $.0063_{\pm.001}$ | $\mathbf{.9941}_{\pm.000}$ | $.9583_{\pm.001}$ | $.2016_{\pm.037}$ | $.1738_{\pm.032}$ |
| $\mathcal{S}$-LRR | $\mathbf{.0359}_{\pm.001}$ | $\mathbf{.0977}_{\pm.004}$ | $\mathbf{.1680}_{\pm.006}$ | $\mathbf{.0062}_{\pm.001}$ | $.9941_{\pm.000}$ | $\mathbf{.9585}_{\pm.001}$ | $.2068_{\pm.035}$ | $.1811_{\pm.031}$ |
| QFD$^2$ | $.0362_{\pm.001}$ | $.0986_{\pm.004}$ | $.1696_{\pm.006}$ | $.0063_{\pm.001}$ | $.9940_{\pm.000}$ | $.9582_{\pm.001}$ | $.1917_{\pm.035}$ | $.1665_{\pm.031}$ |
| $\mathcal{S}$-QFD$^2$ | $\mathbf{.0359}_{\pm.001}$ | $\mathbf{.0977}_{\pm.004}$ | $.1681_{\pm.006}$ | $\mathbf{.0062}_{\pm.001}$ | $.9941_{\pm.000}$ | $\mathbf{.9585}_{\pm.001}$ | $\mathbf{.2086}_{\pm.036}$ | $\mathbf{.1822}_{\pm.032}$ |
| CJS | $.0361_{\pm.001}$ | $.0984_{\pm.004}$ | $.1692_{\pm.006}$ | $.0063_{\pm.001}$ | $.9941_{\pm.000}$ | $.9582_{\pm.001}$ | $.1975_{\pm.040}$ | $.1722_{\pm.035}$ |
| $\mathcal{S}$-CJS | $\mathbf{.0359}_{\pm.001}$ | $.0978_{\pm.004}$ | $.1682_{\pm.006}$ | $\mathbf{.0062}_{\pm.001}$ | $.9941_{\pm.000}$ | $\mathbf{.9585}_{\pm.001}$ | $.2080_{\pm.035}$ | $.1804_{\pm.031}$ |

Table 14: Experimental results of LDL on the `emotion6` dataset formatted as $(\text{mean} \pm \text{std})$

| Algorithms | Cheby. ↓ | Clark ↓ | Can. ↓ | KLD ↓ | Cosine ↑ | Int. ↑ | Spear. ↑ | Ken. ↑ |
|---|---|---|---|---|---|---|---|---|
| LDSVR | $.3152_{\pm.010}$ | $1.8217_{\pm.020}$ | $4.1452_{\pm.064}$ | $1.0744_{\pm.081}$ | $.6906_{\pm.015}$ | $.5773_{\pm.012}$ | $.3915_{\pm.030}$ | $.3235_{\pm.025}$ |
| AA-$k$NN | $.3288_{\pm.011}$ | $1.7116_{\pm.026}$ | $3.8757_{\pm.076}$ | $.9512_{\pm.115}$ | $.6632_{\pm.013}$ | $.5564_{\pm.010}$ | $.2920_{\pm.027}$ | $.2401_{\pm.022}$ |
| LDLFs | $.3120_{\pm.010}$ | $1.6625_{\pm.026}$ | $3.7330_{\pm.075}$ | $.5871_{\pm.024}$ | $.7143_{\pm.011}$ | $.5802_{\pm.010}$ | $.3631_{\pm.029}$ | $.3025_{\pm.024}$ |
| DF-BFGS | $.3026_{\pm.010}$ | $1.6765_{\pm.025}$ | $3.7675_{\pm.071}$ | $.5805_{\pm.026}$ | $.7206_{\pm.013}$ | $.5909_{\pm.010}$ | $.3940_{\pm.027}$ | $.3256_{\pm.022}$ |
| KLD ● | $.3037_{\pm.010}$ | $1.6774_{\pm.025}$ | $3.7729_{\pm.074}$ | $.5863_{\pm.027}$ | $.7191_{\pm.013}$ | $.5897_{\pm.011}$ | $.3959_{\pm.028}$ | $.3259_{\pm.023}$ |
| $\mathcal{S}$-KLD | $.3024_{\pm.010}$ | $1.6548_{\pm.026}$ | $3.6984_{\pm.075}$ | $.5631_{\pm.024}$ | $.7282_{\pm.012}$ | $.5926_{\pm.010}$ | $.4063_{\pm.027}$ | $.3361_{\pm.023}$ |
| SCL ● | $.3020_{\pm.010}$ | $1.6750_{\pm.025}$ | $3.7642_{\pm.073}$ | $.5803_{\pm.027}$ | $.7219_{\pm.013}$ | $.5917_{\pm.011}$ | $.4003_{\pm.028}$ | $.3299_{\pm.023}$ |
| $\mathcal{S}$-SCL | $\mathbf{.3018}_{\pm.010}$ | $1.6554_{\pm.026}$ | $3.6993_{\pm.076}$ | $.5631_{\pm.025}$ | $.7281_{\pm.012}$ | $\mathbf{.5936}_{\pm.010}$ | $\mathbf{.4089}_{\pm.027}$ | $\mathbf{.3383}_{\pm.023}$ |
| LRR ● | $.3030_{\pm.010}$ | $1.6736_{\pm.025}$ | $3.7601_{\pm.073}$ | $.5804_{\pm.026}$ | $.7212_{\pm.013}$ | $.5899_{\pm.010}$ | $.3941_{\pm.027}$ | $.3243_{\pm.023}$ |
| $\mathcal{S}$-LRR | $.3028_{\pm.009}$ | $\mathbf{1.6524}_{\pm.026}$ | $\mathbf{3.6923}_{\pm.074}$ | $\mathbf{.5607}_{\pm.023}$ | $\mathbf{.7299}_{\pm.011}$ | $.5923_{\pm.010}$ | $.4078_{\pm.027}$ | $.3373_{\pm.022}$ |

Table 15: Experimental results of LDL on the `Twitter` dataset formatted as $(\text{mean} \pm \text{std})$

| Algorithms | Cheby. ↓ | Clark ↓ | Can. ↓ | KLD ↓ | Cosine ↑ | Int. ↑ | Spear. ↑ | Ken. ↑ |
|---|---|---|---|---|---|---|---|---|
| LDSVR | $.4236_{\pm.008}$ | $2.6722_{\pm.002}$ | $7.3015_{\pm.009}$ | $5.0018_{\pm.115}$ | $.7627_{\pm.008}$ | $.5761_{\pm.008}$ | $.5237_{\pm.008}$ | $.4246_{\pm.007}$ |
| AA-$k$NN | $.3172_{\pm.004}$ | $\mathbf{2.0142}_{\pm.012}$ | $\mathbf{4.5597}_{\pm.043}$ | $3.1429_{\pm.148}$ | $.7926_{\pm.006}$ | $.6024_{\pm.005}$ | $.5014_{\pm.009}$ | $.4432_{\pm.008}$ |
| LDLFs | $.4035_{\pm.014}$ | $2.5461_{\pm.010}$ | $6.8269_{\pm.040}$ | $1.6884_{\pm.115}$ | $.6756_{\pm.018}$ | $.5318_{\pm.013}$ | $.4164_{\pm.013}$ | $.3349_{\pm.010}$ |
| DF-BFGS | $.2982_{\pm.004}$ | $2.4025_{\pm.005}$ | $6.2416_{\pm.020}$ | $.6304_{\pm.012}$ | $.8250_{\pm.006}$ | $.6220_{\pm.004}$ | $.5467_{\pm.008}$ | $.4454_{\pm.007}$ |
| KLD ○ | $.2966_{\pm.004}$ | $2.4059_{\pm.005}$ | $6.2558_{\pm.020}$ | $.6307_{\pm.013}$ | $.8243_{\pm.006}$ | $.6249_{\pm.005}$ | $.5470_{\pm.008}$ | $.4456_{\pm.007}$ |
| $\mathcal{S}$-KLD | $.2995_{\pm.005}$ | $2.4112_{\pm.005}$ | $6.2883_{\pm.020}$ | $.6491_{\pm.013}$ | $.8203_{\pm.006}$ | $.6205_{\pm.005}$ | $.5384_{\pm.009}$ | $.4385_{\pm.008}$ |
| SCL ● | $.2977_{\pm.004}$ | $2.4028_{\pm.005}$ | $6.2435_{\pm.021}$ | $.6262_{\pm.013}$ | $.8256_{\pm.006}$ | $.6233_{\pm.005}$ | $.5488_{\pm.008}$ | $.4471_{\pm.007}$ |
| $\mathcal{S}$-SCL | $.2940_{\pm.005}$ | $2.4059_{\pm.006}$ | $6.2589_{\pm.023}$ | $.6203_{\pm.013}$ | $.8268_{\pm.006}$ | $.6281_{\pm.006}$ | $.5518_{\pm.008}$ | $.4497_{\pm.007}$ |
| LRR ● | $.2984_{\pm.004}$ | $2.4046_{\pm.005}$ | $6.2525_{\pm.019}$ | $.6351_{\pm.012}$ | $.8232_{\pm.006}$ | $.6220_{\pm.004}$ | $.5443_{\pm.008}$ | $\mathbf{.4636}_{\pm.007}$ |
| $\mathcal{S}$-LRR | $\mathbf{.2937}_{\pm.004}$ | $2.4056_{\pm.005}$ | $6.2589_{\pm.019}$ | $\mathbf{.6189}_{\pm.013}$ | $\mathbf{.8271}_{\pm.006}$ | $\mathbf{.6283}_{\pm.005}$ | $\mathbf{.5519}_{\pm.008}$ | $.4498_{\pm.007}$ |

Table 16: Experimental results of LDL on the `Flickr` dataset formatted as $(\text{mean} \pm \text{std})$

| Algorithms | Cheby. ↓ | Clark ↓ | Can. ↓ | KLD ↓ | Cosine ↑ | Int. ↑ | Spear. ↑ | Ken. ↑ |
|---|---|---|---|---|---|---|---|---|
| LDSVR | $.5174_{\pm.006}$ | $2.6364_{\pm.002}$ | $7.2094_{\pm.011}$ | $5.0366_{\pm.086}$ | $.6636_{\pm.008}$ | $.4683_{\pm.006}$ | $.4622_{\pm.009}$ | $.3811_{\pm.008}$ |
| AA-$k$NN | $.3286_{\pm.005}$ | $\mathbf{2.0685}_{\pm.009}$ | $\mathbf{4.9363}_{\pm.033}$ | $2.2172_{\pm.107}$ | $.7200_{\pm.006}$ | $.5582_{\pm.005}$ | $.4265_{\pm.009}$ | $.3465_{\pm.007}$ |
| LDLFs | $.4051_{\pm.011}$ | $2.4012_{\pm.012}$ | $6.3262_{\pm.050}$ | $1.4274_{\pm.077}$ | $.6073_{\pm.015}$ | $.4822_{\pm.011}$ | $.3478_{\pm.014}$ | $.2847_{\pm.012}$ |
| DF-BFGS | $.3007_{\pm.005}$ | $2.1995_{\pm.007}$ | $5.4900_{\pm.025}$ | $.6309_{\pm.011}$ | $.7801_{\pm.005}$ | $.5979_{\pm.004}$ | $.5102_{\pm.009}$ | $.4226_{\pm.008}$ |
| KLD ○ | $.3015_{\pm.005}$ | $2.2008_{\pm.007}$ | $5.4969_{\pm.025}$ | $.6348_{\pm.012}$ | $.7787_{\pm.005}$ | $.5973_{\pm.004}$ | $.5113_{\pm.009}$ | $.4234_{\pm.008}$ |
| $\mathcal{S}$-KLD | $.3052_{\pm.005}$ | $2.2044_{\pm.007}$ | $5.5222_{\pm.026}$ | $.6485_{\pm.012}$ | $.7720_{\pm.005}$ | $.5926_{\pm.004}$ | $.5030_{\pm.009}$ | $.4166_{\pm.008}$ |
| SCL ● | $.3280_{\pm.013}$ | $2.2986_{\pm.024}$ | $5.9247_{\pm.099}$ | $.8301_{\pm.055}$ | $.7268_{\pm.018}$ | $.5713_{\pm.015}$ | $.4566_{\pm.024}$ | $.3748_{\pm.022}$ |
| $\mathcal{S}$-SCL | $\mathbf{.2929}_{\pm.005}$ | $2.2045_{\pm.007}$ | $5.5289_{\pm.028}$ | $.6113_{\pm.012}$ | $.7862_{\pm.005}$ | $\mathbf{.6070}_{\pm.005}$ | $\mathbf{.5265}_{\pm.008}$ | $\mathbf{.4373}_{\pm.007}$ |
| LRR ● | $.3057_{\pm.005}$ | $2.1969_{\pm.007}$ | $5.4763_{\pm.025}$ | $.6431_{\pm.012}$ | $.7752_{\pm.006}$ | $.5929_{\pm.004}$ | $.5047_{\pm.009}$ | $.4229_{\pm.008}$ |
| $\mathcal{S}$-LRR | $.2938_{\pm.005}$ | $2.2013_{\pm.006}$ | $5.5138_{\pm.023}$ | $\mathbf{.6105}_{\pm.012}$ | $\mathbf{.7864}_{\pm.005}$ | $.6058_{\pm.005}$ | $.5261_{\pm.009}$ | $.4369_{\pm.008}$ |

Table 17: Experimental results of LDL on the `Natural_Scene` dataset formatted as (mean $\pm$ std)

| Algorithms | Cheby. ↓ | Clark ↓ | Can. ↓ | KLD ↓ | Cosine ↑ | Int. ↑ | Spear. ↑ | Ken. ↑ |
|---|---|---|---|---|---|---|---|---|
| LDSVR | .4899$_{\pm.016}$ | 2.0831$_{\pm.025}$ | 5.7724$_{\pm.092}$ | 2.0862$_{\pm.085}$ | .5740$_{\pm.017}$ | .4430$_{\pm.015}$ | .4997$_{\pm.015}$ | .3695$_{\pm.012}$ |
| AA-$k$NN | .3113$_{\pm.014}$ | **1.9066**$_{\pm.034}$ | **4.5413**$_{\pm.110}$ | 1.0874$_{\pm.082}$ | .7113$_{\pm.015}$ | .5636$_{\pm.013}$ | .4921$_{\pm.021}$ | .3518$_{\pm.016}$ |
| LDLFs | .2808$_{\pm.034}$ | 2.4329$_{\pm.024}$ | 6.6027$_{\pm.108}$ | **.6464**$_{\pm.118}$ | .7679$_{\pm.046}$ | .5839$_{\pm.043}$ | .5406$_{\pm.058}$ | .4072$_{\pm.045}$ |
| DF-BFGS | .3074$_{\pm.013}$ | 2.4126$_{\pm.017}$ | 6.5896$_{\pm.072}$ | .7603$_{\pm.033}$ | .7381$_{\pm.013}$ | .5568$_{\pm.011}$ | .5110$_{\pm.017}$ | .3837$_{\pm.013}$ |
| KLD ● | .3201$_{\pm.013}$ | 2.4242$_{\pm.017}$ | 6.6560$_{\pm.070}$ | .8285$_{\pm.044}$ | .7172$_{\pm.015}$ | .5485$_{\pm.011}$ | .4958$_{\pm.016}$ | .3715$_{\pm.012}$ |
| $\mathcal{S}$-KLD | .2743$_{\pm.013}$ | 2.3866$_{\pm.020}$ | 6.4733$_{\pm.077}$ | .6608$_{\pm.039}$ | **.7751**$_{\pm.014}$ | .6133$_{\pm.012}$ | .5592$_{\pm.017}$ | .4221$_{\pm.014}$ |
| SCL ● | .3379$_{\pm.014}$ | 2.4800$_{\pm.018}$ | 6.8659$_{\pm.076}$ | .8867$_{\pm.035}$ | .7014$_{\pm.014}$ | .4801$_{\pm.014}$ | .4109$_{\pm.018}$ | .3025$_{\pm.013}$ |
| $\mathcal{S}$-SCL | **.2733**$_{\pm.013}$ | 2.3734$_{\pm.018}$ | 6.4376$_{\pm.072}$ | .6703$_{\pm.043}$ | .7744$_{\pm.015}$ | .6156$_{\pm.013}$ | .5573$_{\pm.018}$ | .4207$_{\pm.014}$ |
| LRR ● | .3138$_{\pm.013}$ | 2.4469$_{\pm.018}$ | 6.7118$_{\pm.074}$ | .7703$_{\pm.032}$ | .7363$_{\pm.013}$ | .5456$_{\pm.011}$ | .5056$_{\pm.016}$ | .3782$_{\pm.012}$ |
| $\mathcal{S}$-LRR | .2740$_{\pm.018}$ | 2.3461$_{\pm.023}$ | 6.3467$_{\pm.087}$ | .6867$_{\pm.070}$ | .7715$_{\pm.021}$ | **.6199**$_{\pm.017}$ | **.5595**$_{\pm.023}$ | **.4228**$_{\pm.018}$ |

Table 18: Experimental results of LDL on the `Movie` dataset formatted as (mean $\pm$ std)

| Algorithms | Cheby. ↓ | Clark ↓ | Can. ↓ | KLD ↓ | Cosine ↑ | Int. ↑ | Spear. ↑ | Ken. ↑ |
|---|---|---|---|---|---|---|---|---|
| CPNN | .1337$_{\pm.003}$ | .5639$_{\pm.010}$ | 1.0746$_{\pm.020}$ | .1191$_{\pm.005}$ | .9194$_{\pm.003}$ | .8164$_{\pm.004}$ | .6610$_{\pm.013}$ | **.7080**$_{\pm.002}$ |
| AA-$k$NN | .1223$_{\pm.002}$ | .5451$_{\pm.009}$ | 1.0445$_{\pm.018}$ | .1129$_{\pm.004}$ | .9254$_{\pm.003}$ | .8250$_{\pm.003}$ | .6557$_{\pm.011}$ | .5710$_{\pm.010}$ |
| LDLFs | .1172$_{\pm.003}$ | .5233$_{\pm.013}$ | 1.0134$_{\pm.026}$ | .1086$_{\pm.006}$ | .9305$_{\pm.003}$ | .8324$_{\pm.004}$ | .6929$_{\pm.013}$ | .6051$_{\pm.012}$ |
| DF-BFGS | .1210$_{\pm.002}$ | .5282$_{\pm.009}$ | 1.0158$_{\pm.019}$ | .1084$_{\pm.005}$ | .9289$_{\pm.003}$ | .8301$_{\pm.003}$ | .6848$_{\pm.012}$ | .5963$_{\pm.012}$ |
| LRR ● | .1135$_{\pm.002}$ | .5101$_{\pm.009}$ | .9770$_{\pm.018}$ | .0957$_{\pm.004}$ | .9369$_{\pm.002}$ | .8385$_{\pm.003}$ | .7119$_{\pm.011}$ | .6203$_{\pm.011}$ |
| $\mathcal{S}$-LRR | .1125$_{\pm.002}$ | .5086$_{\pm.009}$ | .9717$_{\pm.018}$ | **.0945**$_{\pm.004}$ | **.9376**$_{\pm.002}$ | .8398$_{\pm.003}$ | **.7126**$_{\pm.011}$ | .6227$_{\pm.011}$ |
| QFD$^2$ ● | .1159$_{\pm.002}$ | .5200$_{\pm.009}$ | .9920$_{\pm.018}$ | .0975$_{\pm.004}$ | .9355$_{\pm.002}$ | .8357$_{\pm.003}$ | .7075$_{\pm.011}$ | .6158$_{\pm.011}$ |
| $\mathcal{S}$-QFD$^2$ | **.1123**$_{\pm.002}$ | .5073$_{\pm.009}$ | .9700$_{\pm.018}$ | **.0945**$_{\pm.004}$ | **.9376**$_{\pm.002}$ | **.8401**$_{\pm.003}$ | .7125$_{\pm.011}$ | .6224$_{\pm.011}$ |
| CJS ● | .1153$_{\pm.002}$ | .5127$_{\pm.009}$ | .9845$_{\pm.019}$ | .0984$_{\pm.004}$ | .9352$_{\pm.002}$ | .8368$_{\pm.003}$ | .7103$_{\pm.012}$ | .6178$_{\pm.011}$ |
| $\mathcal{S}$-CJS | **.1123**$_{\pm.002}$ | **.5072**$_{\pm.009}$ | **.9699**$_{\pm.018}$ | **.0945**$_{\pm.004}$ | **.9376**$_{\pm.002}$ | **.8401**$_{\pm.003}$ | .7125$_{\pm.011}$ | .6223$_{\pm.011}$ |

Table 19: Experimental results of LDL on the `fbp5500` dataset formatted as (mean $\pm$ std)

| Algorithms | Cheby. ↓ | Clark ↓ | Can. ↓ | KLD ↓ | Cosine ↑ | Int. ↑ | Spear. ↑ | Ken. ↑ |
|---|---|---|---|---|---|---|---|---|
| CPNN | .1864$_{\pm.005}$ | 1.3367$_{\pm.009}$ | 2.3604$_{\pm.020}$ | .1664$_{\pm.005}$ | .9281$_{\pm.004}$ | .7958$_{\pm.005}$ | .8688$_{\pm.005}$ | .7831$_{\pm.007}$ |
| AA-$k$NN | .1515$_{\pm.004}$ | **1.0443**$_{\pm.015}$ | **1.7295**$_{\pm.031}$ | .1846$_{\pm.016}$ | .9419$_{\pm.004}$ | .8317$_{\pm.005}$ | .8865$_{\pm.006}$ | .8123$_{\pm.008}$ |
| LDLFs | .1307$_{\pm.003}$ | 1.2787$_{\pm.010}$ | 2.1703$_{\pm.024}$ | .1002$_{\pm.005}$ | .9575$_{\pm.003}$ | .8552$_{\pm.004}$ | .9060$_{\pm.005}$ | .8352$_{\pm.007}$ |
| DF-BFGS | .1341$_{\pm.003}$ | 1.2889$_{\pm.010}$ | 2.1982$_{\pm.023}$ | .1050$_{\pm.005}$ | .9551$_{\pm.003}$ | .8523$_{\pm.004}$ | .9047$_{\pm.005}$ | .8337$_{\pm.007}$ |
| LRR | .1312$_{\pm.003}$ | 1.2767$_{\pm.010}$ | 2.1655$_{\pm.024}$ | .1004$_{\pm.004}$ | .9575$_{\pm.002}$ | .8547$_{\pm.003}$ | .9059$_{\pm.004}$ | .8350$_{\pm.006}$ |
| $\mathcal{S}$-LRR | **.1302**$_{\pm.003}$ | 1.2796$_{\pm.010}$ | 2.1717$_{\pm.024}$ | **.0997**$_{\pm.005}$ | **.9576**$_{\pm.002}$ | .8558$_{\pm.003}$ | .9063$_{\pm.004}$ | .8425$_{\pm.006}$ |
| QFD$^2$ ● | .1380$_{\pm.003}$ | 1.2803$_{\pm.010}$ | 2.1858$_{\pm.024}$ | .1084$_{\pm.005}$ | .9535$_{\pm.003}$ | .8476$_{\pm.004}$ | .9021$_{\pm.004}$ | .8297$_{\pm.006}$ |
| $\mathcal{S}$-QFD$^2$ | .1321$_{\pm.004}$ | 1.2811$_{\pm.010}$ | 2.1779$_{\pm.024}$ | .1027$_{\pm.006}$ | .9561$_{\pm.003}$ | .8537$_{\pm.004}$ | .9044$_{\pm.005}$ | .8330$_{\pm.007}$ |
| CJS ● | .1343$_{\pm.003}$ | 1.3057$_{\pm.010}$ | 2.2374$_{\pm.024}$ | .1084$_{\pm.005}$ | .9544$_{\pm.003}$ | .8527$_{\pm.004}$ | .9044$_{\pm.004}$ | .8334$_{\pm.006}$ |
| $\mathcal{S}$-CJS | **.1302**$_{\pm.003}$ | 1.2802$_{\pm.010}$ | 2.1731$_{\pm.024}$ | **.0997**$_{\pm.005}$ | .9575$_{\pm.002}$ | **.8559**$_{\pm.003}$ | **.9066**$_{\pm.004}$ | **.8429**$_{\pm.007}$ |

Table 20: Experimental results of IncomLDL on the `JAFFE` dataset formatted as (mean $\pm$ std)

| Algorithms | $\omega = 20\%$ | | | | | | | |
|---|---|---|---|---|---|---|---|---|
| | Cheby. ↓ | Clark ↓ | Can. ↓ | KLD ↓ | Cosine ↑ | Int. ↑ | Spear. ↑ | Ken. ↑ |
| IncomLDL ● | .0898$_{\pm.010}$ | .3304$_{\pm.024}$ | .6742$_{\pm.049}$ | **.0425**$_{\pm.007}$ | **.9598**$_{\pm.007}$ | .8861$_{\pm.009}$ | .4742$_{\pm.094}$ | .4017$_{\pm.082}$ |
| $\mathcal{S}$-IncomLDL | **.0863**$_{\pm.013}$ | **.3179**$_{\pm.036}$ | **.6525**$_{\pm.077}$ | .0433$_{\pm.012}$ | .9590$_{\pm.011}$ | **.8893**$_{\pm.014}$ | **.5034**$_{\pm.114}$ | **.4401**$_{\pm.100}$ |

| Algorithms | $\omega = 40\%$ | | | | | | | |
|---|---|---|---|---|---|---|---|---|
| | Cheby. ↓ | Clark ↓ | Can. ↓ | KLD ↓ | Cosine ↑ | Int. ↑ | Spear. ↑ | Ken. ↑ |
| IncomLDL ● | .0946$_{\pm.010}$ | .3454$_{\pm.026}$ | .7073$_{\pm.053}$ | .0465$_{\pm.007}$ | .9558$_{\pm.007}$ | .8801$_{\pm.010}$ | .4231$_{\pm.086}$ | .3534$_{\pm.074}$ |
| $\mathcal{S}$-IncomLDL | **.0868**$_{\pm.013}$ | **.3211**$_{\pm.038}$ | **.6568**$_{\pm.081}$ | **.0439**$_{\pm.014}$ | **.9585**$_{\pm.012}$ | **.8886**$_{\pm.015}$ | **.5075**$_{\pm.115}$ | **.4434**$_{\pm.098}$ |

Table 21: Experimental results of IncomLDL on the `SBU_3DFE` dataset formatted as (`mean ± std`)

| Algorithms | $\omega = 20\%$ | | | | | | | |
|---|---|---|---|---|---|---|---|---|
| | Cheby. ↓ | Clark ↓ | Can. ↓ | KLD ↓ | Cosine ↑ | Int. ↑ | Spear. ↑ | Ken. ↑ |
| IncomLDL ● | .1088 $_{\pm.003}$ | .3586 $_{\pm.008}$ | .7586 $_{\pm.017}$ | .0574 $_{\pm.003}$ | .9439 $_{\pm.003}$ | .8655 $_{\pm.003}$ | .3171 $_{\pm.027}$ | .2746 $_{\pm.023}$ |
| $\mathcal{S}$-IncomLDL | **.1014** $_{\pm.004}$ | **.3208** $_{\pm.009}$ | **.6727** $_{\pm.019}$ | **.0516** $_{\pm.003}$ | **.9485** $_{\pm.003}$ | **.8790** $_{\pm.004}$ | **.4255** $_{\pm.026}$ | **.3679** $_{\pm.023}$ |

| Algorithms | $\omega = 40\%$ | | | | | | | |
|---|---|---|---|---|---|---|---|---|
| | Cheby. ↓ | Clark ↓ | Can. ↓ | KLD ↓ | Cosine ↑ | Int. ↑ | Spear. ↑ | Ken. ↑ |
| IncomLDL ● | .1104 $_{\pm.003}$ | .3621 $_{\pm.008}$ | .7673 $_{\pm.017}$ | .0586 $_{\pm.003}$ | .9426 $_{\pm.003}$ | .8638 $_{\pm.003}$ | .3003 $_{\pm.024}$ | .2595 $_{\pm.020}$ |
| $\mathcal{S}$-IncomLDL | **.1016** $_{\pm.004}$ | **.3213** $_{\pm.009}$ | **.6748** $_{\pm.019}$ | **.0516** $_{\pm.003}$ | **.9485** $_{\pm.003}$ | **.8786** $_{\pm.004}$ | **.4232** $_{\pm.025}$ | **.3659** $_{\pm.022}$ |

Table 22: Experimental results of LDL4C on `JAFFE` and `Twitter` formatted as (`mean ± std`)

| Algorithms | JAFFE | | Algorithms | Twitter | |
|---|---|---|---|---|---|
| | 0/1 loss ↓ | Err. prob. ↓ | | 0/1 loss ↓ | Err. prob. ↓ |
| LDL4C ● | .4973 $_{\pm.108}$ | .7665 $_{\pm.020}$ | LDL4C | .9081 $_{\pm.009}$ | .8846 $_{\pm.005}$ |
| $\mathcal{S}$-LDL4C | **.4453** $_{\pm.102}$ | **.7600** $_{\pm.019}$ | $\mathcal{S}$-LDL4C | .8714 $_{\pm.207}$ | .8729 $_{\pm.156}$ |
| LDL-HR | .4786 $_{\pm.097}$ | .7676 $_{\pm.020}$ | LDL-HR ● | .3656 $_{\pm.017}$ | .4928 $_{\pm.011}$ |
| $\mathcal{S}$-HR | .4653 $_{\pm.105}$ | .7655 $_{\pm.019}$ | $\mathcal{S}$-HR | **.2753** $_{\pm.013}$ | **.4250** $_{\pm.008}$ |
| LDLM | .4787 $_{\pm.109}$ | .7687 $_{\pm.021}$ | LDLM ● | .2814 $_{\pm.013}$ | .4291 $_{\pm.008}$ |
| $\mathcal{S}$-LDLM | .4737 $_{\pm.097}$ | .7689 $_{\pm.019}$ | $\mathcal{S}$-LDLM | **.2753** $_{\pm.014}$ | **.4250** $_{\pm.008}$ |

Table 23: Experimental results of LDL4C on `sBU_3DFE` and `Flickr` formatted as (`mean ± std`)

| Algorithms | sBU_3DFE | | Algorithms | Flickr | |
|---|---|---|---|---|---|
| | 0/1 loss ↓ | Err. prob. ↓ | | 0/1 loss ↓ | Err. prob. ↓ |
| LDL4C | .5578 $_{\pm.028}$ | .7671 $_{\pm.007}$ | LDL4C | .8971 $_{\pm.008}$ | .8884 $_{\pm.004}$ |
| $\mathcal{S}$-LDL4C | .5526 $_{\pm.025}$ | .7686 $_{\pm.006}$ | $\mathcal{S}$-LDL4C | .8705 $_{\pm.138}$ | .8702 $_{\pm.100}$ |
| LDL-HR ● | .5167 $_{\pm.027}$ | .7596 $_{\pm.006}$ | LDL-HR ● | .4513 $_{\pm.015}$ | .5823 $_{\pm.007}$ |
| $\mathcal{S}$-HR | .5069 $_{\pm.025}$ | .7598 $_{\pm.006}$ | $\mathcal{S}$-HR | **.4219** $_{\pm.015}$ | **.5639** $_{\pm.007}$ |
| LDLM ● | .5258 $_{\pm.034}$ | .7619 $_{\pm.009}$ | LDLM ● | .4384 $_{\pm.014}$ | .5740 $_{\pm.007}$ |
| $\mathcal{S}$-LDLM | **.4809** $_{\pm.024}$ | **.7524** $_{\pm.005}$ | $\mathcal{S}$-LDLM | .4321 $_{\pm.016}$ | .5667 $_{\pm.007}$ |

Table 24: Experimental results of LE on the `JAFFE` dataset formatted as (`mean ± std`)

| Algorithms | Cheby. ↓ | Clark ↓ | Can. ↓ | KLD ↓ | Cosine ↑ | Int. ↑ |
|---|---|---|---|---|---|---|
| LP | .0812 $_{\pm.001}$ | .3446 $_{\pm.002}$ | .7125 $_{\pm.005}$ | .0424 $_{\pm.001}$ | .9618 $_{\pm.001}$ | .8808 $_{\pm.001}$ |
| GLLE | .0821 $_{\pm.002}$ | .3196 $_{\pm.013}$ | .6518 $_{\pm.028}$ | .0386 $_{\pm.003}$ | .9638 $_{\pm.002}$ | .8901 $_{\pm.004}$ |
| LEVI | .0787 $_{\pm.003}$ | .3316 $_{\pm.013}$ | .6864 $_{\pm.028}$ | .0391 $_{\pm.003}$ | .9649 $_{\pm.002}$ | .8860 $_{\pm.004}$ |
| LIBLE ● | .0813 $_{\pm.006}$ | .3106 $_{\pm.020}$ | .6358 $_{\pm.044}$ | .0370 $_{\pm.005}$ | .9652 $_{\pm.005}$ | .8929 $_{\pm.008}$ |
| $\mathcal{S}$-LIBLE | **.0770** $_{\pm.003}$ | **.2942** $_{\pm.007}$ | **.5997** $_{\pm.016}$ | **.0332** $_{\pm.002}$ | **.9685** $_{\pm.002}$ | **.8987** $_{\pm.003}$ |

Table 25: Experimental results of LE on the `Yeast_heat` dataset formatted as (`mean ± std`)

| Algorithms | Cheby. ↓ | Clark ↓ | Can. ↓ | KLD ↓ | Cosine ↑ | Int. ↑ |
|---|---|---|---|---|---|---|
| LP | **.0421** $_{\pm.000}$ | .2148 $_{\pm.000}$ | .4711 $_{\pm.001}$ | .0153 $_{\pm.000}$ | .9860 $_{\pm.000}$ | .9235 $_{\pm.000}$ |
| GLLE | .0481 $_{\pm.001}$ | .2114 $_{\pm.005}$ | .4282 $_{\pm.011}$ | .0168 $_{\pm.001}$ | .9842 $_{\pm.001}$ | .9298 $_{\pm.002}$ |
| LEVI | .0494 $_{\pm.007}$ | .2125 $_{\pm.027}$ | .4307 $_{\pm.056}$ | .0169 $_{\pm.004}$ | .9838 $_{\pm.004}$ | .9289 $_{\pm.009}$ |
| LIBLE ● | .0453 $_{\pm.000}$ | .1973 $_{\pm.001}$ | .3982 $_{\pm.003}$ | .0148 $_{\pm.000}$ | .9859 $_{\pm.000}$ | .9346 $_{\pm.000}$ |
| $\mathcal{S}$-LIBLE | .0445 $_{\pm.000}$ | **.1901** $_{\pm.002}$ | **.3790** $_{\pm.005}$ | **.0137** $_{\pm.000}$ | **.9869** $_{\pm.000}$ | **.9376** $_{\pm.001}$ |

