# OpenReview forum: "Learning Label Distribution with Subtasks"
_ICLR.cc/2025/Conference — Submitted to ICLR 2025_

### Official Review · Reviewer_v2H9 · 2024-10-30

**Soundness:** 2
**Presentation:** 1
**Contribution:** 2
**Rating:** 3
**Confidence:** 4

**Summary:**

This paper studies the label distribution problem. This paper argues that the recent LDL work seems to exhibit a notable contradiction: 1) some existing LDL methods employ auxiliary tasks to enhance performance, which narrows their focus to specific domains, thereby lacking generalization capability; 2) conversely, LDL methods without auxiliary tasks rely on losses tailored solely to label distributions of the primary task, lacking additional supervised information to guide the learning process. This paper proposes to solve the contradiction by subtasks.

**Strengths:**

The idea of using subtasks in LDL seems to be a novel strategy.

**Weaknesses:**

The representation of this paper is weak. First, this paper does not clearly illustrate the auxiliary task and subtask. Without a clear definition, it isn't easy to get the core idea of this paper. Second, this paper does not explain why the auxiliary tasks will not address the first key issue: satisfying the non-negative and sum-to-one constraints. To my knowledge, those two constraints can easily be satisfied. Just name a few. Those unclear representations render it difficult for the reviewer to understand the core idea of this paper.

The experimental comparison is weak, as most compared methods were published several years ago.

**Questions:**

See the weakness.

**Details Of Ethics Concerns:**

No.

---

> ### Author Response · Authors · 2024-11-15
>
> Many thanks for your precious comments! We have provided point-by-point responses to your questions below.
>
> **Comment 1:** First, this paper does not clearly illustrate the auxiliary task and subtask.
>
> **Response:** We have placed significant emphasis on these terms from the outset of the manuscript to ensure clarity. In **Section 1**, we have clearly emphasized the definition of auxiliary tasks and subtasks. Specifically, auxiliary tasks are those *learned concurrently alongside the primary task*, and subtasks are *serving as auxiliary task*, and they *provide different views of the primary task distribution* in the context of LDL. Furthermore, in **Section 3**, we have provided a problem definition for learning with subtasks. To aid in comprehension, in **Table 1**, we have illustrated key notation and terminology related to subtasks as much as possible.
>
> **Comment 2:** Second, this paper does not explain why the auxiliary tasks will not address the first key issue: satisfying the non-negative and sum-to-one constraints. To my knowledge, those two constraints can easily be satisfied.
>
> **Response:** We would like to clarify that the first key issue is not solely about satisfying the non-negative and sum-to-one constraints, but rather about *fitting* the correct label distribution within the probability simplex space. Many existing LDL methods assume that the label distribution can be represented by a maximum entropy model or by a simple neural network using a softmax function. However, the exponential part of these models restrict the generality of the distribution form, e.g., mixture distributions, sparse distributions, etc.
>
> Auxiliary tasks, while valuable for improving feature representations, do not directly address the challenge of constructing a more flexible and expressive label distribution mixture. In contrast, our $\mathcal{S}$-LDL method focuses on leveraging the reconstructability of subtasks, since subtasks provide different views of the primary task distribution, rendering the mixture of distributions more traceable.
>
> **Comment 3:** The experimental comparison is weak, as most compared methods were published several years ago.
>
> **Response:** We respectfully disagree with the comment that our experimental comparison is weak due to the inclusion of older methods. In fact, the methods we compare against, including DF(*Inf. Fusion*, 2021), SCL(*TKDE*, 2021), HR(*IJCAI*, 2021), LDLM(*ICML*, 2021), LRR(*TKDE*, 2023), QFD$^2$(*ICCV*, 2023), and CJS(*ICCV*, 2023), etc., were published between 2021 and 2023, which makes them highly relevant for current benchmarking.
>
> Moreover, we intentionally include well-established classic methods like SA-BFGS (i.e., KLD) (*TKDE*, 2016) and AA-$k$NN(*TKDE*, 2016), because they still offer valuable insights and remain competitive on certain key metrics. These methods continue to be a benchmark in the field of LDL, and many recent works still compare with them to demonstrate the efficacy of newer approaches [1, 2, 3].
>
> [1] Imbalanced label distribution learning. *AAAI*, 2023.
>
> [2] Predicting label distribution from tie-allowed multi-label ranking. *TPAMI*, 2023.
>
> [3] Adaptive weighted ranking-oriented label distribution learning. *TNNLS*, 2024.

---

### Official Review · Reviewer_VdEY · 2024-10-30

**Soundness:** 3
**Presentation:** 2
**Contribution:** 3
**Rating:** 6
**Confidence:** 3

**Summary:**

The authors proposed a new methodology called S-LDL for label distribution learning problems. The algorithm first constructs subtasks without any extra knowledge and proposes a plug-and-play method framework based on the pseudo-supervised information from subtasks. Experiments demonstrate that S-LDL is effective and efficient.

**Strengths:**

The authors propose a novel method for LDL by partitioning the primary task into multiple subtasks. This method needs no additional auxiliary tasks and provides different views of the primary task distribution, rendering the mixture of distributions more traceable. Moreover, they propose a new aggregation method to be seamlessly compatible with existing LDL methods, and adaptable to derivative tasks of LDL.

**Weaknesses:**

1. Lack of theoretical explanation concerning the proposed subtask construction method.

2. Lack of experiments showing that the S-LDL method is a better learning paradigm than other ensemble methods based on subtasks when both methods use the same subtask construction method.

3. Section 5 " S-LDL OF THE DEEP REGIME" is not very well-written. It should be clarified which variables are obtained by subtask construction and which variables are learned by minimizing Equ. (9).

**Questions:**

1. Can you provide some insights or explanation about Equ.(1)? Why minimizing Equ.(1) can yield diverse subtasks? Is Equ.(1) related to some existing pairwise similarity metrics or is it proposed by yourselves?

2. Could you show the superiority of minimizing Equ. (1) to learn the subtasks over other methods to partition the label space empirically and/or theoretically? In experiments, can you conduct an ablation experiment to compare your Equ. (1) with existing subtask construction methods (e.g., random sampling label space)?

3. Could you compare your method with other subtask-based ensemble methods when both use the same subtask construction method (I.e. minimizing Equ. (1))?

4. If I understand correctly, you train $\phi$, $\psi$, and $\omega$ simultaneously. If I did not, how did you train $\phi$ guided by subtasks separately?

---

> ### Author Response · Authors · 2024-11-15
>
> Many thanks for your precious comments! We have provided point-by-point responses to your questions below.
>
> **Comment 1:** Lack of theoretical explanation concerning the proposed subtask construction method.
>
> **Response:** We appreciate the reviewer’s feedback regarding the theoretical analysis, and we believe that our work has provided a solid foundation for these aspects. In **Section 4.1**, we have discussed the validity of subtask construction in improving performance, highlighting the rationale behind the $\mathcal{S}$-LDL framework. In **Section 4.2**, we have demonstrated that the subtask label distributions can reconstruct the label distribution of the primary task, which further supports the effectiveness of the subtask construction process. Additionally, in **Section 4.3**, we have provided a detailed analysis of the time complexity of subtask construction, establishing its scalability and efficiency. These discussions collectively address the concerns raised about the rationality and optimality of subtask construction, offering a comprehensive justification for our method.
>
> **Comment 2 (Weakness 2 and Question 3):** Lack of experiments showing that the $\mathcal{S}$-LDL method is a better learning paradigm than other ensemble methods based on subtasks when both methods use the same subtask construction method.
>
> **Response:** We appreciate the reviewer’s suggestion. However, we regret that no existing ensemble methods are directly comparable to $\mathcal{S}$-LDL of the shallow regime, since there are currently no other LDL methods utilizing subtask construction technique. Although there are ensemble strategy methods for LDL, none of them works well for incomplete distributions (i.e., subtask label distributions), as highlighted in **Section 2**.
>
> **Comment 3:** Section 5 is not very well-written. It should be clarified which variables are obtained by subtask construction and which variables are learned by minimizing Equ. (9).
>
> **Response:** In Section 5, we have highlighted that the learnable parts of the model are $\varphi$, $\psi$, and $\omega$. Therefore, the model parameters $\boldsymbol{\Theta}$ in Equ. (9) refers to the parameters of all these learnable parts. We will improve the expression in the revised version.
>
> **Comment 4:** Can you provide some insights or explanation about Equ.(1)? Why minimizing Equ.(1) can yield diverse subtasks? Is Equ.(1) related to some existing pairwise similarity metrics or is it proposed by yourselves?
>
> **Response:** The first term in Equ. (1) is designed to avoid unreasonable ignorance by ensuring that the construction of subtask masks depends on the label distribution matrix. This helps to ensure that the subtask label spaces carry more reasonable and relevant information, thus improving the overall task modeling.
>
> The second term is related to pairwise cosine similarity. Specifically, we aim to make the rows of the matrix $\boldsymbol{M}$ (i.e., the subtask masks) as dissimilar as possible. By minimizing this term, we encourage the creation of diverse subtasks, which leads to the generation of more varied and meaningful knowledge across subtasks.
>
> **Comment 5:** Could you show the superiority of minimizing Equ. (1) to learn the subtasks over other methods to partition the label space empirically and/or theoretically? In experiments, can you conduct an ablation experiment to compare your Equ. (1) with existing subtask construction methods (e.g., random sampling label space)?
>
> **Response:** Thank you for the constructive suggestion. We think this is good advice! In fact, as the parameter $\lambda$ increases, the subtask label spaces tend to behave like random sampling. Therefore, the experimental results in Fig. 2(b) provide an insightful comparison. In response to your suggestion, we will provide more detailed experiments in the revised manuscript.
>
> **Comment 6:** If I understand correctly, you train $\varphi$, $\psi$, and $\omega$ simultaneously. If I did not, how did you train $\varphi$ guided by subtasks separately?
>
> **Response:** We train $\varphi$, $\psi$, and $\omega$ simultaneously.

---

> > ### Comment · Reviewer_VdEY · 2024-12-02
> >
> > Thanks for your response. I still have several concerns.
> >
> > 1. The motivation of Eq. (1) is still not well explained in the paper. You need to provide a clearer illustration of the two terms in Eq. (1) in the paper. For example, I suggest you theoretically analyze the resulting subtasks under extreme values of $\lambda$ such as $\lambda \to 0$ and $\lambda \to \infty$ for demonstration.
> >
> > 2. Why not take $\lambda = 0.1$? From Figure 1(a), the optimal value of $\lambda$ seems closer to $0.1$.
> >
> > 3. The authors introduce the definition of information rate and mask valid rate but no theoretical results are established concerning them.

---

> > > ### Author Response · Authors · 2024-12-03
> > > **Thank You**
> > >
> > > Thank you for your thoughtful feedback and continued interest in our work! We appreciate your attention to the theoretical aspect of the paper.
> > >
> > > 1. We have included an illustration in Fig. 2(b), where the effect of varying $\lambda$ on the resulting subtasks is shown. Although the analysis is empirical, it will provide a clearer understanding of the two terms in Eq. (1). $\lambda \to 0$ and $\lambda \to \infty$ are not appropriate for generating subtasks. We will provide a more detailed theoretical analysis to further explore the resulting subtasks once we find a suitable solution.
> > > 2. $\lambda = 0.1$ is also a feasible choice. We have selected $\lambda = 0.2$ because we observe performance improvement in terms of practical results, as demonstrated in Fig. 2(b). We will clarify this to ensure a better understanding.
> > > 3. The concepts of information rate and mask valid rate are introduced for determining hyperparameters without prior knowledge. They have been calculated on all datasets. While we do not provided a full theoretical analysis for these metrics, our intention is to use them merely as a bridge for determining the appropriate ranges for $\lambda$ and $T$.
> > >
> > > Once again, thank you for your valuable comments. We will strive to further improve our manuscript. We appreciate your time and consideration!

---

### Official Review · Reviewer_ZngQ · 2024-11-01

**Soundness:** 2
**Presentation:** 2
**Contribution:** 2
**Rating:** 5
**Confidence:** 4

**Summary:**

This paper proposes S-LDL, which partitions the label distribution of the primary task into subtask label distributions, i.e., a form of pseudo-supervised information, to solve existing problems in LDL.  This paper also conducts several experiments to demonstrate that S-LDL is effective and efficient.

**Strengths:**

1. This work is the first one endeavoring to address LDL via subtasks.
2. This work conducts abundant experiment to show the effectiveness of S-LDL.

**Weaknesses:**

1. There is room for improvement in the structure and content organization of this paper.Expanding the fifth chapter to more clearly elaborate on deep S-LDL could be beneficial. The analysis about subtask construction in the fourth chapter does not seem to serve its intended purpose, and the reasons for this view will be detailed below.
2. The two definitions introduced in Section 4.1, "Validity Analysis," are not formally utilized, and the relevant analysis is conducted solely through experiments, which makes it appear somewhat lacking in rigor.
3. The proof in Section 4.2, "Reconstructability Analysis," seems to yield a distribution that meets the “sum to one” condition, but I do not fully understand why this distribution must necessarily be the original distribution.

**Questions:**

The concept of deep regime S-LDL appears to be quite similar to Error-Correcting Output Codes (ECOC). I would like to know if the subtasks can only be applied within LDL, or if they can be utilized in other contexts as well?

---

> ### Author Response · Authors · 2024-11-15
>
> Many thanks for your precious comments! Responses to your concerns are as follows.
>
> **Comment 1:** There is room for improvement in the structure and content organization of this paper. Expanding the fifth chapter to more clearly elaborate on deep $\mathcal{S}$-LDL could be beneficial.
>
> **Response:** We would like to clarify that the current structure of our manuscript is intentional and is designed to gradually introduce the deep $\mathcal{S}$-LDL framework in a logical progression. We appreciate the reviewer’s suggestion regarding about expanding Section 5. We agree that further elaboration could enhance the clarity of this section. In the revised version of the paper, we will incorporate additional discussions and provide a more comprehensive explanation of the deep $\mathcal{S}$-LDL framework.
>
> **Comment 2:** The two definitions introduced in Section 4.1 are not formally utilized, and the relevant analysis is conducted solely through experiments, which makes it appear somewhat lacking in rigor.
>
> **Response:** The two definitions, information rate and mask valid rate, have been calculated for all datasets with varying $\lambda$, results of which have shown in **Fig. 2 (a)**. Without a specific dataset, the two metrics would be meaningless, as they depend on empirical data to provide relevant insights. Therefore, an empirical analysis, which is conducted through experiments, is more pragmatic in this scenario. We intentionally use $\mathcal{S}$-LDL of the shallow regime to directly assess the pure performance gains brought about by the subtask construction. The purpose of this is to ensure rigor.
>
> **Comment 3:** The proof in Section 4.2 seems to yield a distribution that meets the sum-to-one condition, but I do not fully understand why this distribution must necessarily be the original distribution.
>
> **Response:** The purpose of Section 4.2 is to demonstrate that, under certain conditions, the subtask label distributions can indeed reconstruct the primary task label distribution. In Section 5, we concatenate the subtask label distribution with the representation to serve as input to a label distribution estimator, simulating this reconstruction process. Therefore, it is crucial for the overall validity of the $\mathcal{S}$-LDL framework to prove that the subtask label distributions can yield the original distribution.
>
> **Comment 4:** The concept of deep regime $\mathcal{S}$-LDL appears to be quite similar to Error-Correcting Output Codes (ECOC). I would like to know if the subtasks can only be applied within LDL, or if they can be utilized in other contexts as well?
>
> **Response:** Thank you for your insightful comment! While ECOC decomposes multi-label learning (MLL) problems into multiple binary classification problems, subtasks in $\mathcal{S}$-LDL remain within the LDL framework. These subtasks provide different perspectives on the primary task, allowing for a richer representation and improved performance. We aim for $\mathcal{S}$-LDL to be a straightforward and direct machine learning method, much like ECOC. Moreover, $\mathcal{S}$-LDL can also be extended to other fields, such as classification and label enhancement (LE). In **Section 5**, we discuss how the $\mathcal{S}$-LDL method can be adapted to these fields, highlighting the necessary modifications for effective application outside of LDL.

---

> ### Author Response · Authors · 2024-11-23
>
> Dear Reviewer ZngQ,
>
> We would like to confirm if our responses have addressed your concerns, as the deadline for the discussion process is approaching.
>
> We highly value your feedback and look forward to any further comments or suggestions you may have.

---

### Official Review · Reviewer_ctvs · 2024-11-02

**Soundness:** 3
**Presentation:** 2
**Contribution:** 2
**Rating:** 5
**Confidence:** 4

**Summary:**

This paper proposes a novel Label Distribution Learning (LDL) framework called S-LDL, which solves the problem of label ambiguity by constructing subtasks. S-LDL generates pseudo-supervised information by dividing the label distribution of the main task into subtask label distributions, to enhance the model's generalization ability and ability to utilize additional data. This method does not require additional domain knowledge, can seamlessly integrate with existing LDL methods, and is suitable for LDL-derived tasks.

**Strengths:**

1.This paper proposes a new learning paradigm that enhances LDL by constructing subtasks, an innovative approach that can improve the model's understanding of label distribution.
2.S-LDL does not rely on specific domain knowledge, giving it good generalization ability and allowing it to be applied across different domains.

**Weaknesses:**

1.Although S-LDL reduces reliance on additional training data through the construction of subtasks, the generation and optimization of subtasks may increase the computational burden, especially on large-scale datasets.
2.The performance of S-LDL may be sensitive to parameter selection, such as the number and weight of subtasks, which may require additional adjustments and validation.
3.Although this paper proposes the S-LDL framework, there are some shortcomings in theoretical analysis, especially in the in-depth exploration of the rationality and optimality of subtask construction.

**Questions:**

please see the weaknesses above.

---

> ### Author Response · Authors · 2024-11-15
>
> Many thanks for your precious comments! Responses to your concerns are as follows.
>
> **Comment 1:** Although $\mathcal{S}$-LDL reduces reliance on additional training data through the construction of subtasks, the generation and optimization of subtasks may increase the computational burden, especially on large-scale datasets.
>
> **Response:** It is indeed expected that the training time will increase due to subtasks. However, We would like to emphasize that the $\mathcal{S}$-LDL method achieves significant performance improvements, and the additional computational cost is a reasonable and justifiable trade-off given these gains. As we have highlighted in **Section 4.3**, the overall time complexity of subtask generation is linear with respect to the size of the dataset. This ensures that our method remains scalable and efficient, even when applied to large-scale datasets.
>
> **Comment 2:** The performance of $\mathcal{S}$-LDL may be sensitive to parameter selection, such as the number and weight of subtasks, which may require additional adjustments and validation.
>
> **Response:** The number of subtasks plays an important role in determining performance, which is also indeed expected. As in our reply to the previous comment, the number of subtasks is a crucial hyperparameter that directly affects both performance and computational time. This trade-off is intentional, as the number of subtasks is designed to be adjustable in order to meet the specific needs of different scenarios. We have discussed this in **Section 4.1**, where recommended parameter settings for optimal results are given. As for concerns about other hyperparameters, we perform parameter sensitivity experiments on them in **Section 6**.
>
> **Comment 3:** Although this paper proposes the $\mathcal{S}$-LDL framework, there are some shortcomings in theoretical analysis, especially in the in-depth exploration of the rationality and optimality of subtask construction.
>
> **Response:** We appreciate the reviewer’s feedback regarding the theoretical analysis, and we believe that our work has provided a solid foundation for these aspects. In **Section 4.1**, we have discussed the validity of subtask construction in improving performance, highlighting the rationale behind the $\mathcal{S}$-LDL framework. In **Section 4.2**, we have demonstrated that the subtask label distributions can reconstruct the label distribution of the primary task, which further supports the effectiveness of the subtask construction process. Additionally, in **Section 4.3**, we have provided a detailed analysis of the time complexity of subtask construction, establishing its scalability and efficiency. These discussions collectively address the concerns raised about the rationality and optimality of subtask construction, offering a comprehensive justification for our method.

---

> ### Author Response · Authors · 2024-11-23
>
> Dear Reviewer ctvs,
>
> We would like to confirm if our responses have addressed your concerns, as the deadline for the discussion process is approaching.
>
> We highly value your feedback and look forward to any further comments or suggestions you may have.

---

### Meta-Review · Area_Chair_6nbr · 2024-12-20

**Metareview:**

This paper addresses the contradiction in label distribution learning (LDL) between methods relying on auxiliary tasks and those solely utilizing primary task losses. It proposes $\mathcal{S}$-LDL, a minimalist, generalizable, and plug-and-play framework that generates subtasks as pseudo-supervised information without requiring extra knowledge, making it compatible with existing LDL methods. The paper has received mixed feedback from reviewers. However, three reviewers maintain negative opinions, citing concerns about the paper being difficult to follow, unclear definitions, and insufficient theoretical analysis. Based on these considerations, the paper is not recommended for acceptance at ICLR'25.

**Additional Comments On Reviewer Discussion:**

During the rebuttal period, despite the authors' efforts to address the concerns raised, the opinions of the three reviewers remained unfavorable.

---

### Decision · Program_Chairs · 2025-01-22

Reject